# A Globally Seamless Terrestrial Evapotranspiration Dataset Retrieved by a Nonparametric Approach with Remote Sensing and Reanalysis Datasets

Suyi Liu[a,b,c,†], Xin Pan[b,c,d,†]*, Jie Yuan[a,b,c,†], Kevin Tansey[e], Zi Yang[a,b,c], Zhanchuan Wang[a,b,c], Xu Ding[a,b,c], Yuanbo Liu[f], Yingbao Yang[b,d,c]*

[a]School of Earth Sciences and Engineering, Hohai University, Nanjing 211100, China

[b]Jiangsu Province Engineering Research Center of Water Resources and Environment Assessment using Remote Sensing, Hohai University, Nanjing 211100, China

[c]Key Laboratory of Soil and Water Processes in Watershed, Hohai University, Nanjing 210098, China

[d]College of Geography and Remote Sensing, Hohai University, Nanjing 210098, China

[e]School of Geography, Geology and the Environment, University of Leicester, Leicester LE1 7RH, UK;

[f] Key Laboratory of Lake and Watershed Science for Water Security, Nanjing Institute of Geography and Limnology, Chinese Academy of Sciences, Nanjing 210008, China

*Corresponding author*: Xin Pan, px1013@hhu.edu.cn

Yingbao Yang, yyb@hhu.edu.cn

[†]These authors contributed equally to this work.

**Abstract.** Evapotranspiration (ET) serves as a key indicator of the water change between the Earth's surface and atmosphere, significantly influencing the hydrology cycle, surface energy cycle, and carbon cycle. Existing remote sensing models for estimating ET usually necessitate the parameterization of resistance parameters. In this study, we proposed the Remote Sensed Non-Parametric (RSNP) model, which leverages the nonparametric (NP) and Surface Flux Equilibrium-nonparametric (SFE-NP) approaches, and adapted remote sensing and reanalysis datasets of meteorological and surface parameters as model inputs. We estimate global monthly ET from 2001 to 2019 in the spatial resolution of 0.1° with RSNP model. Validation against FLUXNET sites globally yield RMSE of 23 mm/month (278 mm/yr), while regional-scale validation against water-balance ET results in a Root Mean Square Error (RMSE) of 113 mm/yr. In addition, the produced ET dataset have great accuracy in forest underlying and obtains spatial details of land surface ET. Furthermore, compared with ETMonitor, PEW and PML_V2, our dataset offers a continuous and seamless ET dataset suitable for global research. This study contributes to the advancement of global ET estimation and informs future water balance studies. The dataset

presented in this article has been published in National Tibetan Plateau Data Center at
     https://doi.org/10.11888/Terre.tpdc.301343(Pan, 2024).

**1 Introduction**

       Terrestrial evapotranspiration (ET), consisting of soil evaporation and vegetation transpiration, is
one of the key components in the land-atmosphere water, energy, and carbon cycles, and plays a critical
role in hydrological, metrological, and ecological research(Fisher et al., 2017; Gentine et al., 2019). ET
at the point scale is often observed by some ground observations (e.g. eddy covariance (EC), large-
aperture scintillometer). However, the distribution of these flux sites across the global land surface is
sparse and the coverage period of available data varies at each flux sits, making it difficult to continuously
monitor ET over large areas and conduct simultaneous continuous observations over long time series
through point-scale observations. The ability of conduct periodic and repetitive observations of regions
and cost-effectiveness makes remote sensing capable of conducting global ET observations(Liu et al.,
2022; Zhang et al., 2016). Based on hydrometeorological approaches (e. g. Penman-Monteith (PM)
approach, Priestly-Taylor (PT) approach), various remote sensing models have been proposed
sequentially, such as the Surface Energy Balance System (SEBS), Surface Energy Balance Algorithms
for Land (SEBAL), triangle approach (Bastiaanssen et al., 1998b; Bastiaanssen et al., 1998a; Su, 2002;
Moran et al., 1994) and so on. They have been widely applied to retrieve ET in many regions (Ma et al.,
2013; Singh et al., 2008; Stisen et al., 2008). In addition, in the context of global greening and global
climate change, the study of the regulatory role and response mechanism of the global ET in the global
water and energy cycles has gradually become a key focus in climate research(Yang et al., 2023).

50       Recently, global-scale estimates of ET derived from remote sensing data have been proposed,
including MODIS-MOD16 dataset, Penman–Monteith–Leuning Version 2 (PML_V2) dataset,
Calibration-free (CR) dataset, ETMonitor dataset,  a simplified surface energy-water balance model
based on proportionality hypothesis (PEW) dataset, three temperature (3T) dataset and so on(Mu et al.,
2011). Among them, the spatial and temporal resolutions of global ET datasets varied from 500 m to 1°
and from daily to annual, and they have been globally validated with a relative mean square error (RMSE)
value ranges from 26.03 mm/month to 35.36 mm/month(Elnashar et al., 2021), and have been used in

global studies(Cheng et al., 2020; Ma and Zhang, 2022; Zheng et al., 2019). Although there are several global ET remote sensing datasets available. They often have limitations in practical application due to gaps caused by cloudy conditions and the desert regions(Chen et al., 2021). In addition, limited by the

complex parametrization of resistances and the empirical determination of coefficients in those remote sensing ET models, the applicability and accuracy of them have not been incrementally improved, especially in the studies of hydrology, metrology, and ecology. It is necessary to provide a reliable remote sensing dataset of global terrestrial seamless ET based on a non-empirical/physical approach or model.

The Nonparametric (NP) method and Surface Flux Equilibrium- Nonparametric  method (SFE-NP)

based on Hamilton's principle and relative humidity budget, and avoids the complex parametrization of resistances and the empirical determination of coefficients(Liu et al., 2012; Pan et al., 2024). The validation of NP and SFE-NP method at various EC sites represented the RMSE at daily resolution was 11-34 W/m$^2$, and both showed a relatively satisfactory performance of ET estimation at the point scale around the world(Pan et al., 2024). Related models of remote sensing based on NP and SFE-NP

approaches have been built and successfully applied to the retrieval of regional ET in Heihe River basin, Poyang Lake basin, and Mekong River Basin(Liu et al., 2022; Pan et al., 2023; Pan et al., 2022) respectively. To expand the applicability of those models, a globally improved model based on NP method (namely RSNP model), is proposed in this paper, from which a global, seamless ET dataset has been produced. Evaluation using data from EC sites and at a basin scale, and we also discussed

comparative analysis between our dataset and other datasets.

## 2 Data

### 2.1 Model Forcing Data

Remote sensing data and reanalysis datasets are used as the input data of the RSNP model to estimate ET at a global scale during 2001-2019. The monthly surface albedo and Broadband Emissivity

(BBE) are from Global Land Surface Satellite (GLASS) in a spatial resolution of 0.05° (Zhao et al., 2013). The monthly air temperature, land surface temperature, surface thermal radiation downwards, surface solar radiation downwards, and air pressure are from the fifth generation of European Reanalysis –Land (ERA5-Land) in a spatial resolution of 0.1°(Muñoz-Sabater et al., 2021). The Moderate-



resolution Imaging Spectroradiometer (MODIS) Land Cover Type (MCD12Q1) Version 6.1 data,

supplied by the National Aeronautics and Space Administration (NASA), provided IGBP land cover

types in a resolution of 1 km  and was used to support the estimation of soil heat flux(Sulla-Menashe et

al., 2019). All remote sensing product and reanalysis datasets were resampled to the spatial resolution of

0.1°×0.1° before being adapted to the RSNP model.

**Table 1 Remote sensing and reanalysis datasets used in the RSNP model**

| Dataset | Variables | Spatial resolution | Temporal resolution | Data Usage |
|---|---|---|---|---|
| GLASS | Black sky Albedo<br>White sky Albedo<br>Broadband Emissivity (BBE) | 0.05°×0.05° | 8-day | ET Retrieval |
| ERA5-Land | Skin temperature<br>Surface pressure<br>Surface solar radiation downwards<br>Surface thermal radiation downwards<br>2m Temperature<br>2m Dew point temperature | 0.1°×0.1° | Monthly | ET Retrieval |
| The water-balance-based ET on dataset of large river basins of the world | Water-balance-based evapotranspiration data | | Annual | ET Validation |
| MCD12Q1 | Land cover type | 1 km×1 km | Annual | ET Retrieval |
| Version 3 of the Global Aridity Index and Potential Evapotranspiration Database | Arid Index | 1 km×1 km | | ET Validation |

**2.2 Flux Tower Data**

At the point scale, ground observations of ET from the FLUXNET2015 Dataset

(https://fluxnet.org/data/fluxnet2015-dataset/) were used to validate and access the accuracy of monthly



ET retrieved by remote sensing method. Under the standard of energy closure rates between 0.8 and 1.0

and at least five consecutive months of valid data, 88 globally distributed sites were selected with ten

different types of underlying surfaces, including MF (Mixed Forest), GRA (Grassland), SAV (Savanna),

WSA (Woody Savanna), EBF (Evergreen Broadleaf Forest), CRO (Cropland), DBF (Deciduous

Broadleaf Forest), ENF (Evergreen Needleleaf Forest), WET (Wetland), OSH (Open Shrublands).

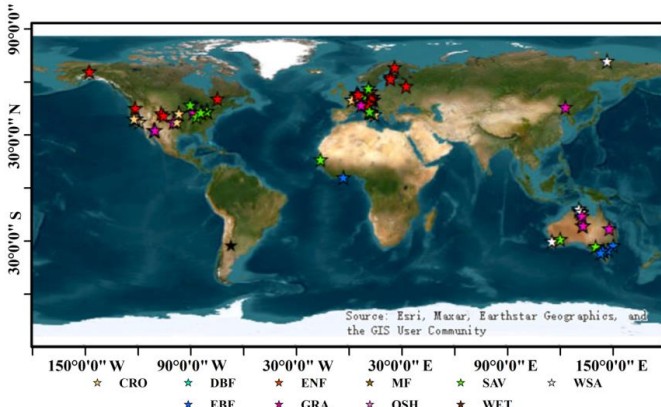

**Figure 1: The distribution of FLUXNET sites used in this study**

**2.3 Water Balance Validation Data**

The annual water-balance-based ET (*ETwb*) dataset in the worldwide large river basins during

1983-2016 (Ma et al., 2024) was used as water balance validation data in this study. This dataset was

derived from National Tibetan Plateau Data Center (https://data.tpdc.ac.cn/en/data). We excluded basins

that cover less than $2\times10^5$ km$^2$ and ultimately selected 38 basins, and the distribution is shown in Fig. 2.

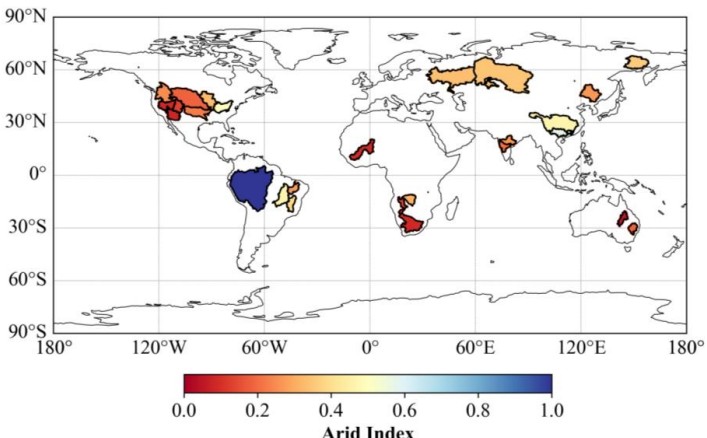

**Figure 2: The geographical distribution and the Arid Index of 38 basins used in this study. (Basin area cover more than $2\times10^5$ km$^2$)**

**2.4 Other Global ET Datasets Used for Cross Validation**

In this study, three existing global ET datasets, ETMonitor, PML_V2, and PEW were selected to cross-validate the global ET results of the RSNP model. ETMonitor is based on the Shuttleworth-Wallace dual-source model, the improved Gash model, and Penman's equation for different underlying surfaces to estimate ET(Zheng et al., 2022) PML_V2 is based on the Penman-Monteith-Leuning (PML) model (Zhang et al., 2019); PEW is constructed based on the PT-JPL algorithm to estimate ET based on a surface energy-water balance framework(Fu et al., 2022). To unify the spatial scales of these datasets, ETMonitor and PML_V2 were resampled to a spatial resolution consistent with that of RSNP ET by using the nearest-image resampling method, and then the differences in the simulation of global land surface ET by different remote sensing models were explored in terms of the time series and spatial distribution, respectively. Among them, the ET in arid desert steppes and desert areas often amount to nearly zero or is missing(Xiao et al., 2024).

**3 Methodology**

**3.1 Global Nonparametric Evapotranspiration Models**

Based on the Hamilton of a microstate system, LE is expressed in the NP approach as Equation (1-1)(Liu et al., 2012). However, NP approach has shown good accuracy in remote sensing applications in



**Table 2 Other global terrestrial ET datasets used for cross-validation**

| ET Datasets | Method | Spatial resolution | Temporal resolution | Time span | Reference |
|---|---|---|---|---|---|
| ETMonitor | Estimating ET components with a multi-process parameterization model | 1 km×1 km | Monthly | 2003-2018 | (Zheng et al., 2022) |
| PML_V2 | PML model coupled with gross primary products via canopy conductance theory | 0.5 km×0.5 km | Daily | 2002-2019 | (Zhang et al., 2019) |
| PEW | PT-JPL algorithm considering available water capacity | 0.1°×0.1° | Monthly | 1982-2018 | (Fu et al., 2022) |

wet regions, but its accuracy at the site scale is limited in arid regions(Hsieh et al., 2022; Yang et al., 2016). To expand the applicability of the original NP approach, the SFE-NP approach is proposed to estimate ET in a water-limited situation(Pan et al., 2024). In the SFE-NP approach, LE can be expressed as Equation (1-2).

$$LE_{NP} = \frac{\Delta}{\Delta+\gamma}(R_n - G_s) - \varepsilon_s\sigma\left(T_s^{\,4} - T_a^{\,4}\right) + G_s l\,n\left(\frac{T_s}{T_a}\right), \tag{1-1}$$

$$LE_{SFE-NP} = \frac{RH\Delta}{RH\Delta+\gamma}(R_n - G_s) - \varepsilon_s\sigma\left(T_s^{\,4} - T_a^{\,4}\right) + G_s l\,n\left(\frac{T_s}{T_a}\right), \tag{1-2}$$

where $R_n$ is the total surface net radiation, $G_s$ is the soil heat flux, $\varepsilon_s$ is the surface emissivity (derived from GLASS), $\sigma$ is the Stephan-Boltzmann constant, $T_s$ is the land surface temperature (LST) (derived from ERA5-Land), $T_a$ is the air temperature (derived from GLASS), $\Delta$ is the slope of the saturated vapor pressure at temperature $T_a$, $\gamma$ is the psychometric constant, and $\sigma$ is the Stefan-Boltzmann constant ($\sigma =$

$5.67 \times 10^{-8} W/(m^2 k^4)$), $RH$ is the relative humidity. In this study, for the global terrestrial ET, the NP method was adapted to humid areas, and the SFE-NP method was adapted to arid areas.

Furthermore, $R_n$ can be expressed as(Bisht et al., 2005):

$$R_n = (1-\alpha)R_{sd} + R_{ld} - \varepsilon_s\sigma T_s, \tag{2}$$





where $R_{sd}$ is the surface shortwave downward radiation (derived from ERA5-Land), $R_{ld}$ is the surface

longwave downward radiation (derived from ERA5-Land), and $\alpha$ is the surface albedo (derived from

GLASS).

$G_s$ can be estimated by the Global Land Evaporation Amsterdam Model (GLEAM) method

as(Miralles et al., 2011):

$$G_s = \begin{cases} 0.05 R_n & bare\ soil \\ 0.20 R_n & short\ vegetation, \\ 0.25 R_n & tall\ canopy \end{cases} \tag{3}$$

The *RH* can be estimated as:

$$RH = \frac{e_s(T_d)}{e_s(T_a)}, \tag{4}$$

where $T_d$ is the dew point temperature (derived from ERA5-Land), and $e_s$ means the saturated water vapor

pressure and can be expressed as:

$$e_s(T_d) = \frac{exp[17.62(T_d) - 273.15]}{243.12 + T_d - 273.15}\ , \tag{5-1}$$

$$e_s(T_a) = \frac{exp[17.62(T_a) - 273.15]}{243.12 + T_a - 273.15}\ , \tag{5-2}$$

**3.2 Framework of Global Seamless ET Estimation**

The consistency of spatial and temporal resolution of inputs ensures the computability of multi-

source remote sensing data. In this study, GLASS provided albedo and BBE with 0.05°×0.05° and 8-day

resolution. They were aggregated into monthly images, and then resampled to 0.05°×0.05° by using the

nearest neighbor resampling.

All images used as model inputs have a consistent spatial and temporal resolution and were

seamless across the global land surface (water bodies, and permanent ice and snow were excluded). For

the estimation of surface net radiation, ERA5-Land provided monthly LST, surface thermal radiation

downward, and surface solar radiation downward, GLASS provided the albedo and BBE. For the

estimation of soil heat flux, this study adopted the landcover type provided by MCD12Q1 to classify

different ratios of soil heat flux of short vegetation, high canopy, and bare soil. For the RSNP model,

according to the arid index provided by the Version 3 of the Global Aridity Index and Potential

Evapotranspiration Database, ET in the arid region (where the arid index is less than 0.65) was estimated

with the SFE-NP method, while ET in the wet region (where the arid index is more than 0.65) was

estimated with the NP method.

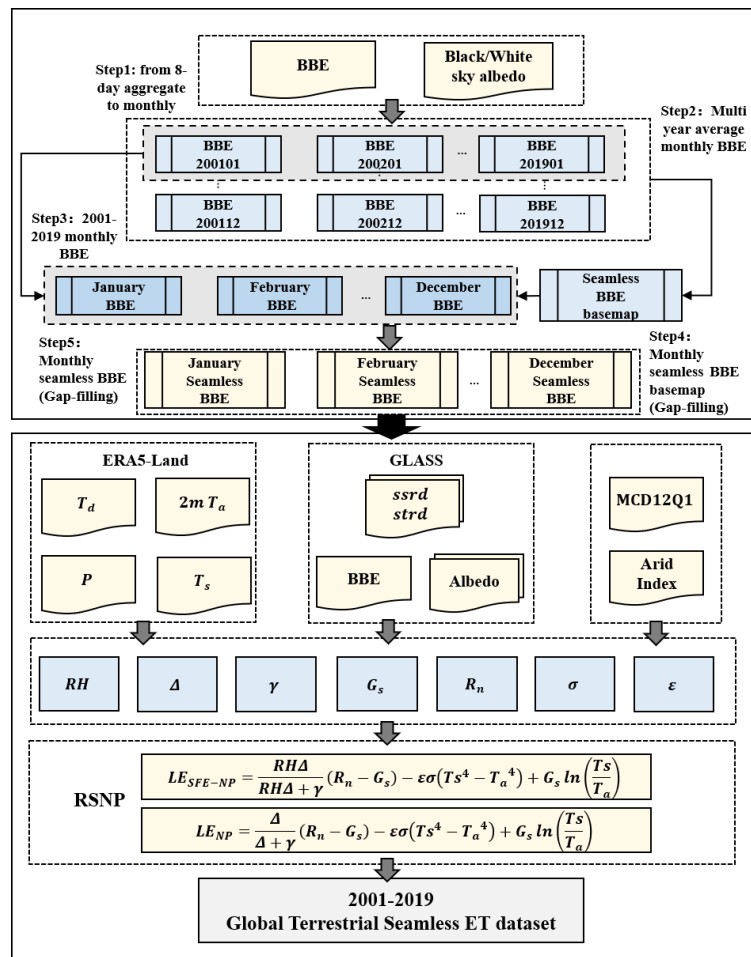

**Figure 3: The data preprocessing of the RSNP remote sensing model and its global retrieval model.**

However, persistent discrepancies in default pixel values for monthly BBE, such as in 2014 with

regional gaps observe in South Africa, Asia, and Australia, posed challenges in achieving a consistent

estimation of ET under these circumstances. to address the absence of pixel values in the monthly BBE

images, a multi-step approach was employed. Initially, a baseline BBE map was generated by averaging

a comprehensive dataset comprising a total of 218 number BBE images. Subsequently, monthly BBE

data for the period 2001-2019 were acquired for each month. Some individual months and regions still

exhibited unavailable pixels in the monthly BBE map. To rectify this issue, the next step was to fill each

month's BBE map with the average monthly BBE of total months. Finally, the gap-filled monthly BBE

maps were used to fill the 218 original monthly BBE images with the corresponding pixels.

### 3.3 Validation Method and Accuracy Metrics

To evaluate the retrieved ET comprehensively, direct validation is taken for the accuracy

evaluation, and cross-validation is analyzed to reveal the discrepancies among different ET datasets in

this study.  In detail, direct validation is composed of validation at the point scale (validated by the LE

observed by EC sites at the monthly scale) and validation at the basin scale (validated by *ETwb* at the

annual scale). For cross-validation, the spatio-temporal discrepancies among different ET remote sensing

datasets were revealed.

In addition, the mean bias error (bias), relative error (RE), and Relative Mean Squared Error

(RMSE) and Correlation Coefficient ($R^2$) were used to reveal the performance of ET estimations  (Jia et

al., 2012).

### 4 Evaluation of ET estimates

### 4.1 Validation and Comparison of Monthly ET with In-Situ Data

Figure 4-1 shows the scatter plot of RSNP retrieval monthly ET and flux tower observed ET over

88 flux tower sites. RSNP exhibited correlation with observed ET from the flux tower data with an $R^2$

value of 0.66, which is consistent with ETMonitor and PML_V2. Over these sites, RSNP displayed great

accuracy with RMSE value of 23.2 mm/month and a bias value of -3.86 mm/month. As a result, the

accuracy of RSNP is comparable to the current level of accuracy of published applications for global ET

datasets. Overall, RSNP has a more concentrated scatter density distribution than the other ET products,

especially when the observed ET from flux tower were higher than 100 mm/month.

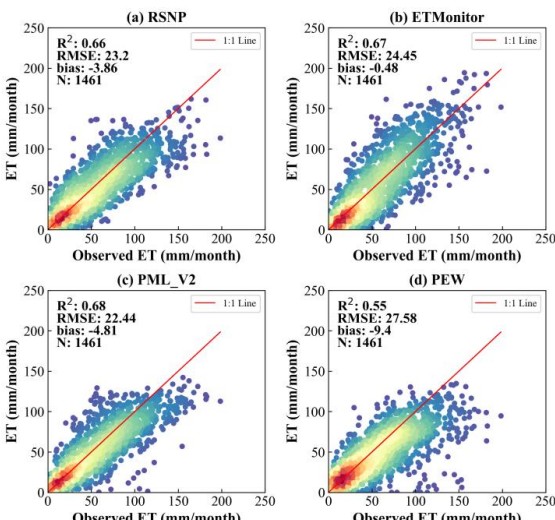

**Figure 4: Comparison of the retrieval ET of each product and observations over 88 FLUXNET sites. The relative mean square error (RMSE) and the bias are both in mm/month.**

Over 88 FLUXNET sites, the performance of RSNP exhibited variations across different land cover and geographical locations, with notable differences observed between continents. The RSNP ET and flux tower observed ET showed great correlation at WET sites, with $R^2$ value is 0.86, followed by that were MF sites, with $R^2$ value is 0.79, except OSH sites with $R^2$ value of 0.32, $R^2$ values of RSNP at each land cover was higher than 0.58, showing the RSNP correlated well at vegetation areas. In terms of precision indicators, the RMSE value was between 13.19 mm/month (MF) and 28.37 mm/month (DBF) over those land covers, and were often comparable or lower than those of other products, indicating its effectiveness in minimizing prediction errors. In terms of the performance of bias, the absolute valud of bias for RSNP model did not exceed 13.34 mm/month. It was slightly overestimated in DBF sites, and conversely underestimated in other sites.Combining the information from Fig. 5, it can be observed that there is significant variability in the retreival accuracy of each product for SAV, EBF and WET, and RSNP model still demonstrate relatively lower retreival errors and higher consistency with the observed ET from flux towers.

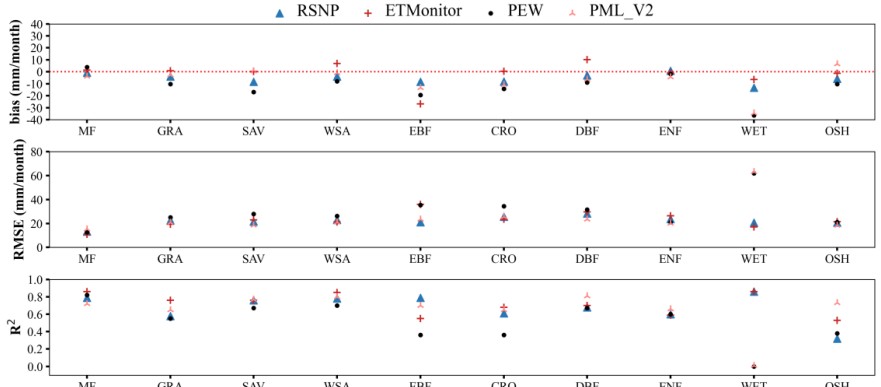

**Figure 5: Comparison of the $R^2$ and RMSE between RSNP retrieval ET and observations of ET over 88 FLUXNET sites at 10 types of land cover including MF (Mixed Forest), GRA (Grassland),**
**SAV (Savanna), WSA (Woody Savanna), EBF (Evergreen Broadleaf Forest), CRO (Cropland), DBF (Deciduous Broadleaf Forest), ENF (Evergreen Needleleaf Forest), WET (Wetland), OSH (Open Shrublands). The relative mean square error (RMSE) is in mm/month.**

### 4.2 Validation and Comparison of Monthly ET with Water Balanced ET

As shown in Fig. 6, RSNP demonstrates a high degree of consistency with the three common ET
products in terms of distribution and accuracy at the basin scale and almost remained the same consistency and accuracy of WBET compared with other ET products (the $R^2$ value was 0.89, RMSE value was 113.04 mm/yr, and RE value was 22%), the RMSE value of RSNP was lower than that of ETMonitor and PEW, and the $R^2$ value of RSNP was also slightly higher than that of ETMonitor and PEW  This study also calculated the average arid index for each basin based on the Version 3 of the
Global Aridity Index and Potential Evapotranspiration Database (Zomer et al., 2022), and compared the accuracy performance of ET datasets at the basin scale. As shown in Fig. 7, the basin RMSE of RSNP and each ET dataset was almost below 200 mm/yr, except for PEW. When arid index is over 1.0, the basin RMSE of RSNP was about 100 mm/yr, while other ET datasets was over 150 mm/yr. RSNP well have certain advantages in monitoring basin or regional ET on a global scale.

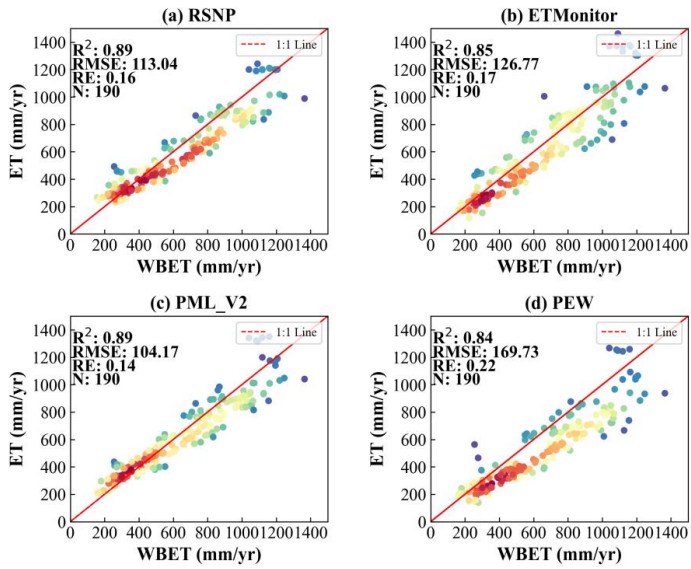

**Figure 6: Comparison of R2, RMSE, RE between the retrieval ET of each dataset and WBET over 38 basins.**

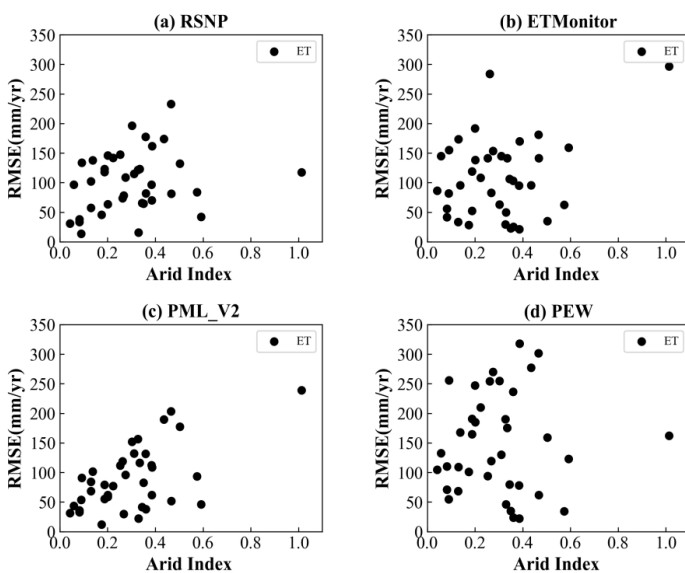

**Figure 7: Distribution of RMSE and Arid Index of basins at regional scale. The black dots display each basin's RMSE and Arid Index; the Arid Index range from low to high represents from arid to humid.**

### 4.3 Cross-comparison of Global ET

### 4.3.1 Temporal Pattern of Monthly ET Datasets

Fig. 8 shows the temporal trends of monthly ET for 4 global ET datasets from 2003-2018. Monthly

ET gradually increased from February to July, with all datasets reaching their peak value in July. The

monthly ET monitored by RSNP fall at an intermediate level among these datasets. During December,

January, and February, the monthly ET were slightly lower than PML_V2 but higher than ETMonitor

and PEW. From March to November, RSNP exhibited relatively stable variations in monthly ET, closely

aligning with ETMonitor. The most pronounced differences in monthly ET among various datasets were

observed especially in July: PML_V2 (64.91 mm/month) > RSNP (59.69 mm/month) > ETMonitor

(59.08 mm/month) > PEW (54.88 mm/month).

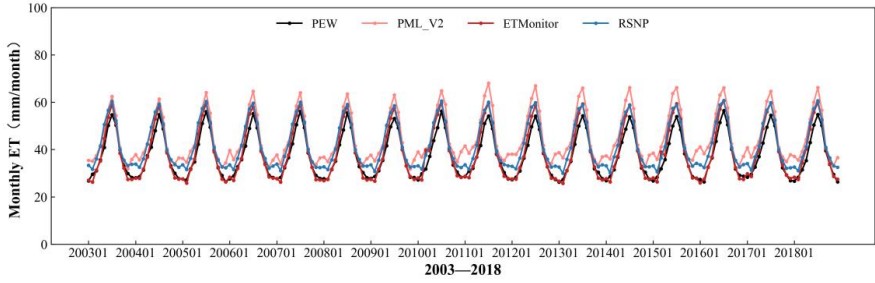

**Figure 8: Global monthly average ET (mm/month) of RSNP, ETMonitor, PML_V2 and PEW during 2003-2018.**

**4.3.2 Spatial Pattern of Annual ET Datasets**

Fig. 9 displays the spatial distribution of annual average ET from 2003 to 2018. Overall, RSNP

shows good agreement with the global spatial distribution of terrestrial ET with other published global

datasets. Fig. 9(a)-(d) shows the global spatial distribution of ET from 2003-2018 of RSNP, ETMonitor,

PML_V2, and PEW. RSNP correlated well and exhibited a high degree of consistency with other datasets.

Specifically, regions with higher ET values globally are observed in the tropical rainforest areas of South

America, Africa, and Indonesia. The RSNP estimated ET in tropical rainforest area to be in the range of

1300-1500 mm/yr, closely resembling the ET value of PML_V2, and slightly lower than that of

ETMonitor, and higher than PEW. However, the magnitudes of ET in these three regions vary across

different datasets. For instance, RSNP indicated the highest ET in central Africa, whereas, on ETMonitor,

the highest ET is observed in northern South America. These discrepancies highlight the inconsistent

performance of ET across various datasets. Apart from Greenland and South America, ET was the lowest

in northern Africa, the Qinghai-Tibet Plateau, and the desert regions of South America. However, many

models set ET to zero during the parameterization process for desert regions, resulting in monthly values

of 0 mm/month, and only RSNP captures the spatial differences in annual ET in the regions of northern

Africa. While RSNP detected annual average ET in African deserts ranging from 0 to 200 mm/yr, other

datasets showed the annual average ET not exceed 50mm/yr. Therefore, from 2003 to 2018, RSNP

slightly overestimated ET in the Africa, with an annual average ET of 590.68 mm/yr, while other datasets

ranged from 448.28 mm/yr to 554.41 mm/yr. In Fig. 9(e), the annual ET exhibits a decreasing trend with

increasing latitude. In both the Northern and Southern Hemispheres, the peak value of RSNP was

observed near 0°latitude, and then annual ET decreases with increasing latitude until near 30°latitude.

ET in the Southern Hemispheres was slightly higher than that in the Northern Hemisphere. According to

the pattern of increasing latitude, RSNP closely matches the average level of latitude-based ET in the

Southern Hemisphere. In terms of the Northern Hemisphere, from 15°N to 30°N latitude, RSNP's annual

ET significantly surpasses that of other datasets, but all ET datasets exhibit a consistent trend with latitude.

As latitude continues to increase, between 35°N-75°N, the differences in latitude-based annual ET

between RSNP and other datasets decreased, nearly aligning with the average level. ETMonitor displayed

the widest range of annual ET changes with latitude, with the lowest estimated ET values in high-latitude

regions (75.70 mm/yr) and the highest values near the equator (1335.40 mm/yr). PEW's latitude-based

annual ET was generally below the average level of the four global ET datasets.

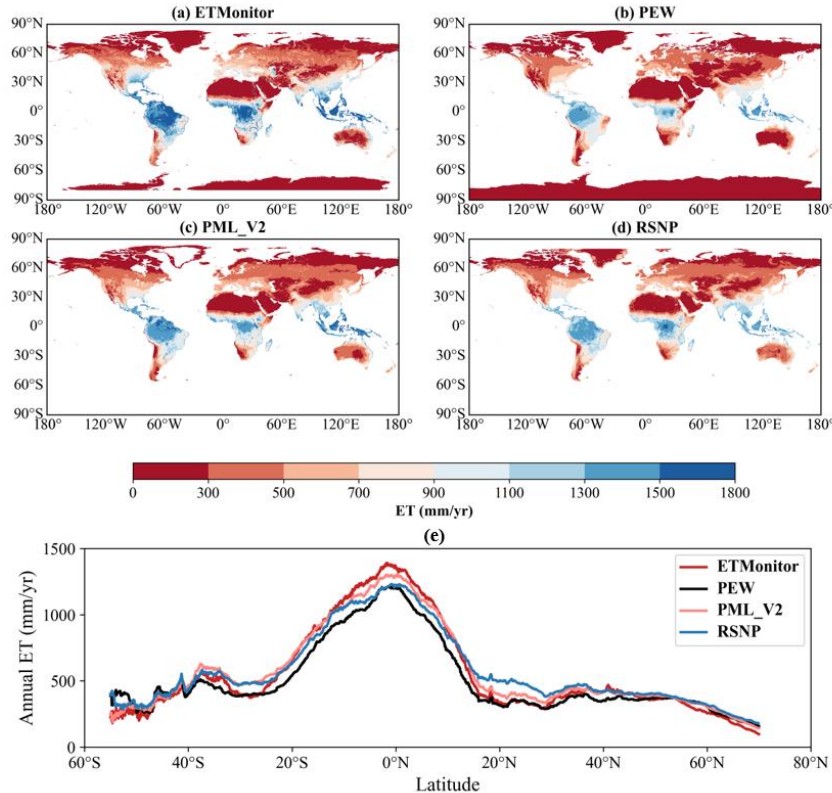

**Figure 9: (a)-(d) The spatial distribution of annual ET during 2003-2018 and (e) the variation of latitude annual ET during 2003-2018.**

## 5 Discussion

### 5.1 The Potential Reasons and Influence of the Data Seam for Other ET Datasets

Although the accuracy of remotely sensed ET datasets is generally acceptable at regional scales, continuous access to daily or monthly ET is not often available for individual pixels that would impact the spatio-temporal continuity required at global scale. In this study, we counted the available pixels ratio of global terrestrial ET images at the monthly scale for RSNP, PEW, ETMonitor, and PML_V2 (pixels in water and permanent snow and ice regions were excluded). As shown in Fig. 10, ETMonitor, PML_V2

and PEW exhibit missing pixels for months at the global scale. ETMonitor and PML_V2 exhibit the lowest monthly pixel availability in January. From May to December, PML_V2 maintained a ratio higher than 85%, and ETMonitor sustains a ratio exceeding 85% from April to December. In addition, it seems

that the pixel availability of ETMonitor and PML_V2 datasets increases as surface radiation levels in the

Northern Hemisphere rise (mainly located at the region with a latitude of large than 30°N) (Fig. 11).

Furthermore, the relatively high ratios of missing pixels in ETMonitor and PML_V2 in some desert

regions (e. g. Sahara desert, Taklimakan desert) The missing pixels in those regions is possibly related

to the insufficient good-quality points available for interpolation (Zhang et al., 2019).

Barren/deserts and middle-high latitude regions account for about 24% in the North hemisphere

and  81% of the total terrestrial area on the Earth's surface, respectively(Mu et al., 2011).  Chen et al.

had verified that the loss of pixels would like to underestimate ET across a global scale. Consequently,

in these regions, the relatively high proportion of missing pixels could compromise the reliability of

global water resources assessments(Tang et al., 2024). Furthermore, the water-energy-carbon nexus in

these regions is highly susceptible to climate variability(Park et al., 2020) incomplete data may make it

crucial to have comprehensive data to ensure a precise understanding of ecological, environmental,

meteorological, and hydrological shifts.  .

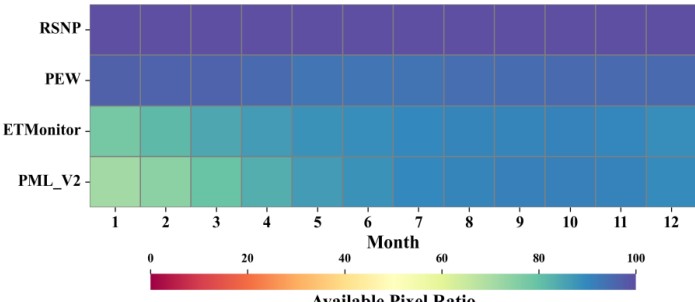

**Figure 10: The available pixel ratio of ET datasets at the monthly scale**

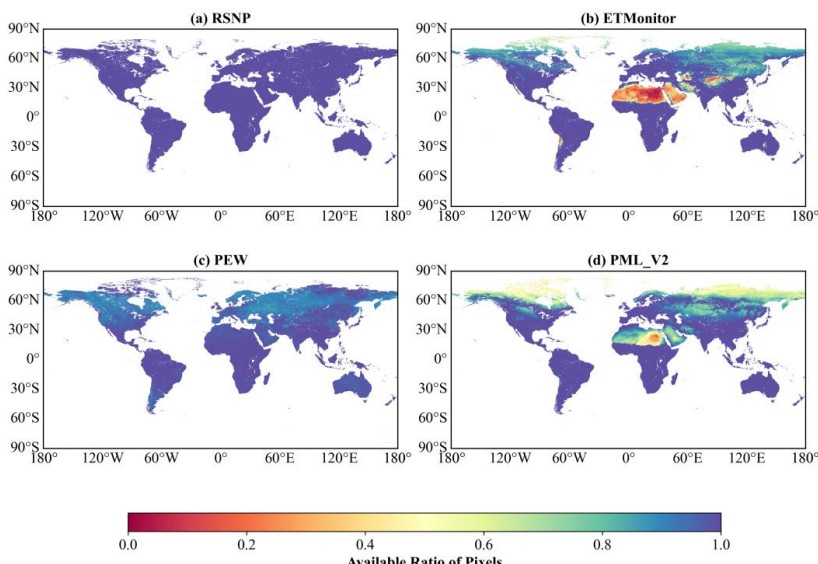

**Figure 11: Monthly available ratio of pixels and the spatial distribution for ET datasets (water, and permanent ice and snow were excluded).**

### 5.2 The Advantages of Our ET Estimations

Many ET remote sensing datasets have been published in recent years (e. g. ETMonitor, PEW, and PML_V2), yet there are numerous locations of missing data, both reflected in the temporal discontinuity (Fig. 10) and spatial discontinuity (Fig.11) of available pixels on a global scale. Considered about the urgent requirement of the tempo-spatial continuous ET in the ecological, hydrological and meteorological studies at the regional and global scale(Ma et al., 2022; Ma and Zhang, 2022), our ET estimation can provide the seamless ET remote sensing dataset at the global scale.

In addition, as the core of the proposed RSNP model, NP approach and its improvement (SFE-NP) can eliminate the uncertainty caused by some empirical parameters (such as surface resistance, vegetation resistance and air resistance) in some traditional approaches (e. g. the PM approach) existed in many ET remote sensing datasets (e. g. ETMonitor, PML_V2 and PEW)(Liu et al., 2012; Pan et al., 2024). Even though the calibration based on the worldwide EC observations can help the determination of those parameters for the global ET retrieval, the accuracy of models and datasets are possibly limited in the region with sparse/no EC sites(Ma et al., 2021). Therefore, the RSNP model and its related datasets of

global ET may be globally reliable without dependence of empirical parameters on calibration or parameterization, especially in some wild regions. For example, in some wild regions of the western China, western North American and western South American (Fig. 12), our datasets can also provide abundant details of ET compared with ETMonitor (spatial resolution: 1km), PML_V2 (spatial resolution: 0.5 km) and PEW (spatial resolution: 0.1°) even though the spatial resolution of our dataset is only 0.1°.

It implies that the spatial resolution is not the only dominant factor for the unsatisfactory performance by these ET datasets. Similar phenomena were also discovered in some studies(Stisen et al., 2008; Zheng et al., 2022).

In addition, the RS-NP model estimates global seamless ET distinct from present global ET datasets. In the recent years, researches of globally terrestrial water-energy budget commonly use the

composition of many globally ET datasets to eliminate the possible error in single dataset (Pan et al., 2020; Yang et al., 2023). However, most parts of those datasets are based on similar principle or method of ET estimation (e. g. PM equation, PT equation), and might have a similar systematic uncertainty. That means the reliability of those researches of globally terrestrial water-energy budget might be affected. Therefore, our model and dataset is helpful for the elimination of the uncertainty in the researches of

globally terrestrial water-energy budget.

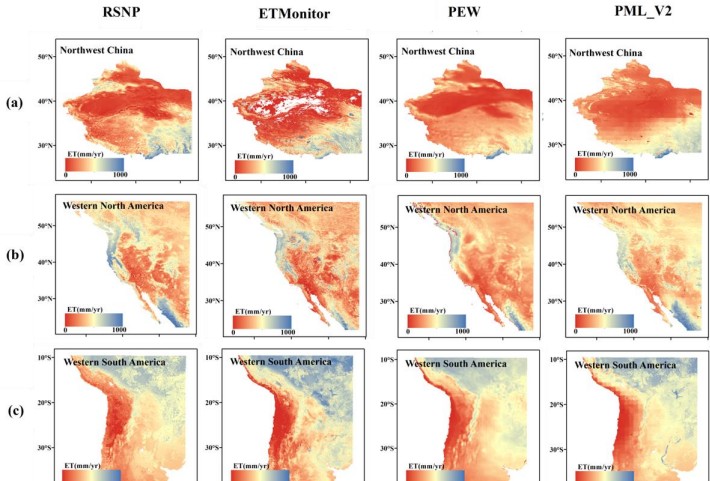

**Figure 12: 4 ET datasets variations in typical regions in 2014. Columns from left to right: (a) Northwest China; (b) Western North America; (c) Western South America. The bottom panels showed the land cover in these regions (land cover data from MCD12Q1).**



**6 Conclusion**

In response to the challenges and complexities associated with the parameterization of land surface characteristics in existing global models, this study introduces a nonparametric global ET model that eliminates the need for the pre-correlation of input data and model parameters. Utilizing skin temperature, surface pressure, surface solar radiation downwards, surface thermal radiation downwards, 2m temperature, 2m dew point temperature provided by ERA5-Land, and albedo and BBE data provided by GLASS, the RSNP model estimated global land surface net radiation. Subsequently, we employed the methodology from the GLEAM model to estimate the soil heat flux. Using these remote sensing data, net radiation and soil heat flux as inputs, the model applied SFE-NP and NP method in arid and humid regions, respectively, facilitating the estimation of global terrestrial actual monthly ET (during 2001-2019) at the spatial resolution of 0.1°.

The validation results showed great accuracy both at point and regional scale. In terms of the validation with FLUXNET sites at the point scale, RSNP showed a RMSE value of 23.2 mm/month, a bias value of -3.86 mm/month and the $R^2$ value of 0.66. Regarding the performance of different underlying surface, RSNP was slightly overestimated in DBF sites, and conversely underestimated in other sites. While According to the comparison between WBET and estimated annual ET, RSNP model displayed a great correlation with the RMSE value of 113.04 mm/yr, RE value of 22%, and $R^2$ value of 0.89.

Comparing the annual ET estimates of RSNP with other ET datasets spanning from 2003 to 2018, there is a high consistency in spatial and temporal resolution characteristics. RSNP closely approximates ETMonitor in capturing the temporal trends of monthly ET. In the spatial distribution cross-validation of annual ET, RSNP reproduces similar spatial ET patterns and latitude-dependent ET trends as current ET datasets. However, the average level of annual ET varies among different ET datasets. In tropical rainforest areas, RSNP closely aligns with PML_V2, and presenting lower values compared to ETMonitor but higher than PEW. Conversely, in desert areas, RSNP captures ET levels slightly higher than existing datasets, showcasing regional variations in ET within desert regions. Furthermore, owing to the interpolation of emissivity data, RSNP's monthly ET from 2001-2019 comprehensively convers the global terrestrial surface, which refers as a seamless ET dataset.

The globally seamless ET dataset estimated by RS-NP model is a different kind of ET datasets with a different principle/method of ET estimation compared with the currently global ET datasets, and

is helpful for the elimination of the systemic uncertainty in the studies of land surface water and energy

cycles at the global scale.

**7 Data Availability**

The seamless global ET data of RSNP model is freely available at National Tibetan Plateau Data Center : https://doi.org/10.11888/Terre.tpdc.301343(Pan, 2024).

**Author Contributions**

Suyi Liu: Conceptualization, Methodology, Resources, Formal analysis, Data Curation, Validation, Writing-Original Draft, Visualization; Xin Pan: Conceptualization, Methodology, Resources, Writing-Reviewing and Editing, Funding acquisition; Yuan Jie: Formal analysis, Visualization, Validation; Kevin Tansey: Writing-Reviewing and Editing, Funding acquisition; Zi Yang: Data Curation; Zhanchuan Wang:

Formal analysis; Xu Ding: Formal analysis; Yuanbo Liu: Supervision, Funding acquisition; Yingbao Yang: Suggestions, Funding acquisition.

**Funding**

This work was supported by the National Nature Science Foundation of China [41701487, 42230112, 42071346 and 42371397], Royal Society IEC\NSFC\223292 - International Exchanges 2022

Cost Share (NSFC) grant of the UK,and Jiangsu Marine Science and Technology Innovation Project [JSZRHYKJ202302].

**Conflicts of Interest**

The authors declare no conflict of interest.



**Acknowledgements**

We thank the FLUXNET community council for providing field observation data (http://www.fluxdata.org), the Climate Data Store (https://cds. climate.copernicus.eu/) for providing ERA5-Land data, the Figshare Open Repository (https://figshare.com/) for providing Aridity Index Database v3 (Global-AI_PET_v3), LP DAAC (https://ladsweb.modaps.eosdis.nasa.gov/) for providing land cover data, Beijing Normal University Data Center (http://glass-dataset.bnu.edu.cn/) for providing

GLASS dataset, the National Tibetan Plateau Data Center (https://data.tpdc.ac.cn/) for providing ETMonitor, PEW, PML-V2, and ETwb datasets.

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
