# Peer review of "A Globally Seamless Terrestrial Evapotranspiration Dataset Retrieved by a Nonparametric Approach with Remote Sensing and Reanalysis Datasets"

_Earth System Science Data, 2024_

## Author Comment (AC1)

**Response to Anonymous Referee #1:**

Dear reviewer, we sincerely appreciate your time and effort in reviewing our manuscript and providing valuable feedback to help improve our work. In the reply, the reviewer's comments are in black, our responses are in blue, and quotes from the revised manuscript are in *orange italics*.

**General Comment**

This paper describes a "globally seamless ET dataset...with remote sensing and reanalysis data". The data is openly available at the National Tibetan Plateau Data Center for the period 2001-2019. I believe that the paper requires a severe revision of its content, as it lacks most of the details behind the methodology adopted, details on the characteristics of the final products are difficult to find (is it monthly or daily? Is the model applied directly on monthly data or aggregated afterward?), and several key details on the validation are difficult to follow. In addition, numerous typos and unclear sentences can be found throughout the text (see examples in the specific comments below).

However, the most notable drawback of the dataset resides in its inception. The study claims that this is a "remote sensing based" dataset that "overcome the need for pre-defined parameters". Regarding the first point, I found really difficult to see this as a remote sensing product, as the vast majority of the inputs came from ERA5-land. The remote sensing contribution is limited to emissivity and albedo only. This would not have been a major issue (beside the need to reword some of the model descriptions), but it highlights the second major problem of this dataset. The method uses skin LST from ERA5-land. These data are not observed from satellite, but they are modelled within the land surface component of the reanalysis system. It means that skin temperature depends on the parameterization used in ERA5-Land, the same parameterization that you are claiming to avoid. Following this consideration, the relationships used (1-1 and 1-2) acts only as a simplified version of the PM approach, where the skin temperature is derived from the more complex (and heavily parameterized) H-TESSEL.

Overall, the dataset may still have some useful applications related to multi-model assessment, but three key points need to be address: 1) a much better contextualization of the modelling framework and scope in view of the above-mentioned issue; 2) a much better description of the methodology, including differences from already existing approaches, and 3) an improved (especially in consistency) evaluation of the dataset against other similar products (e.g., ET from ERA5-land itself).

**Response:** We sincerely thank you for providing such professional and constructive feedback on our manuscript. I deeply appreciate your comments which are invaluable in helping refine and strengthen our research. We have responded to your key comments and each specific comments individually.

The responses to general comments are as follows:

(1) The temporal scale of our model and dataset: The RSNP model estimate global monthly ET dataset with monthly inputs, and we have added corelated information where the dataset first mentioned in both Abstract and section '2.2 Framework of Global ET Estimation and Dataset':

**a.** Abstract: "In this study, we improved the Remote Sensed Non-Parametric (RSNP) model based on the NP and SFE-NP method with remote sensing and reanalysis data, and estimated global monthly ET from 2001 to 2019 in the spatial resolution of 0.1°." (Line 21-23)

b. Section 2.2: "Remote sensing data and reanalysis datasets with full coverage across the global land surface are used as the inputs of the RSNP model to estimate global land surface ET at monthly scale during 2001-2019..." (Line 128-130)

c. Section 2.2: "GLASS 8-day albedo and BBE data were firstly aggregated into monthly scale, ..." (Line 153)

(2) Description of the dataset: The RSNP ET dataset integrates both remote sensing and reanalysis data as inputs, with skin LST obtained from ERA5-Land, rather than direct satellite observations. In response to your comments, we have revised relevant descriptions to remove the term "remote sensing" to avoid any potential misunderstanding. We appreciate your insightful observations, which have helped improve the accuracy of our manuscript.

(3) Modelling of framework: We have reorganized the main structure of our manuscript. In the revised section '2.2 Framework of Global ET Estimation and Dataset', we present the framework together with the introduction of input data and their pre-processing. We also introduce the sequential steps involved in parameter derivation and estimation, ensuring a clear and logical flow of method introduction.

(4) Description of the methodology: We have revised the 'Introduction' and 'Methodology' with a much clearer description of the NP approach.

a. In the last paragraph of Introduction, we extend the basic thesis of the NP method and its development to SFE-NP method: "Evaporation is the phase change process where water molecules transition from liquid to vapor, and thermal driving is the primary mechanism governing terrestrial evaporation. Hamilton's principle offers a physical insight into the macro-state processes to mechanics and describe thermodynamics. The original nonparametric (NP) method is based on the Hamiltonian principle that terrestrial ET follows in the macroscopic state, with surface temperature as a generalized coordinate of the Hamiltonian, and combining with the equilibrium ET (Liu et al., 2012), the original NP method is in a simple analytical form without parameterization of aerodynamic resistances. To address NP method's applicability in arid areas, the surface flux equilibrium (SFE) with relative humidity was introduced to develop the SFE-NP method (Pan et al., 2024)." (Line 74-82)

b. In the section '2.1 Nonparametric approach for global ET Estimation', we have expanded on the basic principles, limitations of the NP method, and the basic principles of the SFE-NP method. We introduced the methods in more detail to enhance the readability of the paper.

(5) Comparison with ERA5-Land ET dataset: According to your suggestion, we acknowledge the importance of a more consistent and rigorous evaluation of the RSNP ET dataset, particularly in comparison to ERA5-Land (which has similar parameters). We have enhanced dataset evaluation by incorporating ERA5-Land monthly ET as a reference dataset, expanding the validation analysis to include accuracy assessment and spatial distribution comparisons.

a. Accuracy comparison: At in-situ EC sites, RSNP ( $R^2=0.65$ , RMSE=23.19 mm/month, bias=-3.81 mm/month) shows consistency with ERA5-Land ( $R^2=0.64$ , RMSE=23.94 mm/month, bias=-1.36 mm/month), and RSNP shows less underestimation than ERA5-Land. For different land cover, *"RSNP shows higher in-situ accuracy than ERA5-Land for vegetated land covers, and the accuracy improvement at the wetland sites is significant, with RMSE reducing from 65.21 mm/month to 20.6 mm/month*" (249-254). At the basin scale, RSNP ( $R^2=0.89$ , RMSE=113.04 mm/month, RE=0.16) explicit a slightly higher error than ERA5-Land ( $R^2=0.89$ , RMSE=101.02 mm/month, RE=0.13), but has the least overestimation among high ET basins (which higher than 1200 mm/yr). b. Spatial distribution comparison: ERA5-Land provide global gap-free ET dataset, however, RSNP demonstrates superior capability of ET variations. For example, in the South America, ERA5-Land shows minimal spatial differentiation in ET among vegetation areas, while RSNP successfully captures the gradual transition of ET variation, better reflecting the actual surface heterogeneity. (383-385)

Further details regarding to these modifications can be found in response to specific comments.

**Specific comments**

**Title: A seamless global...**

**Response:** Thank you very much for your thoughtful review. We have corrected the grammar error, and we have revised "Seamless" to "Gap-free" to describe the dataset's continuous and uninterrupted characteristics. The revised title is "A Gap-free Global Terrestrial Evapotranspiration Dataset Estimated by the Nonparametric Approach with Remote Sensing and Reanalysis Datasets".

**L17. Water exchange**

**Response:** Thank you very much for your thoughtful review. We have corrected the phrase error. (Line 17)

**L18-19. Hydrological, surface energy, and carbon cycles.**

**Response:** Thank you for your thoughtful review. We have checked and corrected the grammatical error of this sentence. (Line 18)

L20. Resistances. This is difficult to follow out of context into an abstract. Please reword. **Response:** Thank you for your thoughtful review. We have revised this sentence to indicate the aerodynamic resistances:

"Different parameterization schemes of aerodynamic resistances might result in uncertainties in global ET dataset." (Line 19)

**L27. Explain acronyms.**

**Response:** We sincerely appreciate your valuable comments. We have revised this sentence, and the name of each ET dataset were removed in the Abstract section. Moreover, we have corrected all acronyms at their first appearance in the manuscript. Thank you again for your thoughtful review.

**L40. To conduct.**

**Response:** Thank you for your thoughtful review. We apologize for the error in grammar, and we have revised this sentence as "*The ability to perform periodic and repetitive observations over regions, coupled with its cost-effectiveness, enables remote sensing conducting global ET observation*". (Line 43-45)

**L41. Conducting. Repetition**

**Response:** Thank you for your thoughtful review. We have rewritten this sentence to avoid the repetition used of "conduct". The revised sentence is:

"The ability to perform periodic and repetitive observations over regions, coupled with its costeffectiveness, enables remote sensing conducting global ET observation". (Line 43-45)

**L44. Sequentially?**

**Response:** Thank you for your careful review. The word "sequentially" may confuse the meaning of the sentence, and we have revised this statement for clarity:

"Some remote sensing models have been proposed based on hydrometeorological approaches (e. g. Penman-Monteith (PM) approach, Priestly-Taylor (PT) approach), such as the Surface Energy Balance System (SEBS), Surface Energy Balance Algorithms for Land (SEBAL), triangle approach (Bastiaanssen et al., 1998b; Bastiaanssen et al., 1998a; Su, 2002; Moran et al., 1994) and so on. They have been widely applied to retrieve ET in many regions". (Line 45-50)

**L50-53. Add references to these datasets.**

**Response:** We sincerely appreciate your feedback on the statement. We had added refences to each global ET dataset to make sure the standardized of science writing.

"Manny global ET datasets derived from remote sensing data and meteorological forcing data have been proposed, including MODIS-MOD16 dataset (Mu et al., 2011), Penman–Monteith–Leuning Version 2 (PML-V2) dataset (Zhang et al., 2019), the Operational Simplified Surface Energy Balance (SSEBop) dataset (Senay et al., 2020), Calibration-free (CR) dataset (Ma et al., 2021), ETMonitor dataset (Zheng et al., 2022), a simplified surface energy-water balance model based on proportionality hypothesis (PEW) dataset (Fu et al., 2022), three temperature (3T) dataset (Yu et al., 2022) and so on." (Line 54-59)

**Reference:**

- Fu, J., Wang, W., Shao, Q., Xing, W., Cao, M., Wei, J., Chen, Z., and Nie, W.: Improved global evapotranspiration estimates using proportionality hypothesis-based water balance constraints, Remote Sens. Environ., 279, 113140, http://doi.org/10.1016/j.rse.2022.113140, 2022.
- Ma, N., Szilagyi, J., and Zhang, Y.: Calibration free complementary relationship estimates terrestrial evapotranspiration globally, Water Resour. Res., 57, e2021WR029691, https://doi.org/10.1029/2021WR029691, 2021.
- Mu, Q., Zhao, M., and Running, S. W.: Improvements to a MODIS global terrestrial evapotranspiration algorithm, Remote Sens. Environ., 115, 1781-1800, http://doi.org/10.1016/j.rse.2011.02.019, 2011.
- Senay, G. B., Kagone, S., and Velpuri, N. M.: Operational global actual evapotranspiration: Development, evaluation, and dissemination, Sensors, 20, 1915, https://doi.org/10.3390/s20071915, 2020.
- Yu, L., Qiu, G. Y., Yan, C., Zhao, W., Zou, Z., Ding, J., Qin, L., and Xiong, Y.: A global terrestrial evapotranspiration product based on the three-temperature model with fewer input parameters and no calibration requirement, Earth Syst. Sci. Data Discuss., 2022, 1-33, https://doi.org/10.5194/essd-14-3673-2022, 2022.
- Zhang, Y., Kong, D., Gan, R., Chiew, F. H., McVicar, T. R., Zhang, Q., and Yang, Y.: Coupled estimation of 500 m and 8-day resolution global evapotranspiration and gross primary production in 2002–2017, Remote Sens. Environ., 222, 165-182, http://doi.org/10.1016/j.rse.2018.12.031, 2019.
- Zheng, C., Jia, L., and Hu, G.: Global land surface evapotranspiration monitoring by ETMonitor model driven by multi-source satellite earth observations, J. Hydrol., 613, 128444,

https://doi.org/10.1016/j.jhydrol.2022.128444, 2022.

**L54. Use consistent units for pixel size.**

**Response:** Thank you for your valuable feedback regarding pixel size units. To ensure clarity and consistency, we have converted  $1^{\circ}$  to approximately 111 km near the equator. Because the actual geographical distance corresponding to 1 radian varies with the geographical location, so we have standardized the spatial resolution description as follows:

"Among them, the spatial and temporal resolutions of global ET datasets varied from 500 meters to almost 1° (approximately 111 kilometers near the equator)..." (Line 60-61)

**L58. Datasets available, they often...**

**Response:** We sincerely appreciate your careful review, and we feel sorry for the writing error. In the revised version, we have removed the original sentence and revised this part to better connect our research focus. The revised statement is, "*Existing global ET datasets still leave room for improvement, primarily relate to the complex parametrization of resistances and the empirical determination of coefficients in global ET models, which affect the applicability and accuracy in the studies of hydrology, meteorology, and ecology. In addition, pixel gaps might lead to limitations in practical application (Chen et al., 2021)." (Line 65-69)*

**L62. Metrology?**

**Response:** Thank you for your careful review. We have corrected the word as "meteorology". (Line 67)

L62-63. In this sentence, it is not clear which problem (or problems) this dataset is trying to solve. **Response:** Thank you for your insightful comments. We apologize for that the original text did not sufficiently highlight the specific scientific gaps addressed by our dataset. To better indicate our research objectives, we have revised the statement with reference to explicitly state that this study: *"By evaluating 25 global ET datasets with site observations and their spatial patterns, Tang et.al refer that ET dataset produced based on similar algorithms tend to have high consistency in annual magnitude and spatial distribution (Tang et al., 2024). Therefore, developing a global ET dataset based on well-defined physical mechanisms remains a critical need in ET research." (Line 69-74)*

**Reference:**

Tang, R., Peng, Z., Liu, M., Li, Z.-L., Jiang, Y., Hu, Y., Huang, L., Wang, Y., Wang, J., and Jia, L. J. R. S. o. E.: Spatial-temporal patterns of land surface evapotranspiration from global products, 304, 114066, https://doi.org/10.1016/j.rse.2024.114066, 2024.

L66. A lot of repetitions (non-parametric) and unclarified terms (what is the role of Hamilton's principle here).

**Response:** We sincerely thank you for your constructive comments. Your suggestions have helped us better introduce the ET estimation method. We have removed redundant mentions of "non-parametric" to improve readability. And we have also revised the statement to indicate the role of Hamilton's principle:

"Hamilton's principle offers a physical insight into the macro-state processes. The original nonparametric (NP) method is based on the Hamiltonian principle that terrestrial ET follows in the macroscopic state, with surface temperature as a generalized coordinate of the Hamiltonian, and combining with the equilibrium ET (Liu et al., 2012), the original NP method is in a simple analytical form without parameterization of aerodynamic resistances." (Line 76-81)

**L66-69. This sentence is unclear, please reword.**

**Response:** Thank you for highlighting the need for greater clarity in this statement. We have revised this sentence to introduce to more clearly present the performance of NP approaches:

"The evaluation of the NP and SFE-NP methods, compared with observed LE across EC sites, reveals that the RMSE ranges from approximately 11 to 34 W/m2 at the daily scale (Pan et al., 2024)." (Line 84-85)

L76. I suggest introducing the methodology first, as explaining the data used, without introducing for what they are used for, make difficult to follow.

**Response:** Thank you for your valuable suggestion and it helps us better organize the readability of this manuscript. We have reorganized Section2 to introduce "Methodology and Materials" with global ET model, model input dataset, and model validation dataset. As suggested, we have initially introduced the methodology, and then introduced the framework of RSNP model together with model inputs, finally introduced the validation datasets. The structure of Section2 is as follows:

2.Methodology and Materials

- 2.1 Nonparametric approach for global ET Estimation
- 2.2 Framework of Global ET Estimation and Model Forcing Data
- 2.3 Datasets for Evaluation
  - 2.3.1EC Observations from FLUXNET2015
  - 2.3.2Water-balance-based ET of Global Basins
  - 2.3.3Other Global ET Datasets

L78. As the inputs of

**Response:** Thank you for your careful review. We have corrected the grammatical error of this sentence. (Line 129)

**L79. To estimate ET.... Daily? Monthly? Not clear.**

**Response:** We sincerely appreciate your valuable suggestion regarding to the temporal resolution of the RSNP ET estimation. The temporal resolution of the ET dataset has been added in the revised sentence, and it locates in Section2.2:

"Remote sensing data and reanalysis datasets are used as the inputs of the RSNP model to estimate global land surface ET at monthly scale during 2001-2019". (Line 128-130)

**L80. At a spatial resolution**

**Response:** Thank you for your careful review, we apologize for the incorrect use of prepositions, and this statement have been revised according to the reorganization of Section2.

L81. Longwave radiation.

**Response:** Thank you for your careful review. We have corrected the term "surface thermal radiation downwards" to "longwave radiation" throughout the manuscript.

L87. Resempled... how? Especially land use, which is categorical.

**Response:** Thank you for your comments on the preprocessing of model inputs. In this paper, the GLASS data were resampled to a spatial resolution of  $0.1^{\circ}$  using the nearest neighbor resampling, and the land use type were aggregated to  $0.1^{\circ}$  by the maximum fraction method. We have modified the pre-processing of model inputs at section2.2:

"The consistency of spatial and temporal resolution in the input images is crucial for ensuring the computability of multi-source remote sensing data. GLASS 8-day albedo and BBE were firstly aggregated into monthly albedo and BBE, considering there were still missing pixels of albedo and BBE which would result in missing value of ET result, the nearest day gap-filling method were used to directly fill the missing pixels at the monthly scale. And finally, the gap-filled global albedo and BBE, and global aridity index datasets were resampled to  $0.1^{\circ}$  using the nearest neighbor resampling. The land use type was aggregated to  $0.1^{\circ}$  by the maximum fraction method before adapted into the RSNP model." (Line 152-158)

Table 1. This table is not referenced in the text, as far as I can tell.

**Response:** We sincerely appreciate your feedback. In the revised version, we added the reference of Table 1 at the beginning of scetion2.2, where these datasets were first mentioned in our manuscript. (Line 131)

Table 1. Please clarify which input is from remote sensing and which from reanalysis. Also, please separate the model inputs from other data used for validation and analysis. The "data usage" column may not be seen by a reader, especially when things are mixed (2 retrievals, then validation, then retrieval again, ...).

**Response:** We sincerely appreciate your insightful suggestions for improving Table 1. We have added a column of "Data Type" to distinguish which input is from remote sensing and which from reanalysis. And according to the change of the manuscript's structure, the revised Table 1 only introduces model input data. The revised Table 1 is as follows:

| Dataset   | Data Type              | Variables                                                                                                                                               | Spatial resolution | Temporal resolution |
|-----------|------------------------|---------------------------------------------------------------------------------------------------------------------------------------------------------|--------------------|---------------------|
| GLASS     | Remote
sensing data | Black sky Albedo
White sky Albedo
Broadband Emissivity (BBE)                                                                                      | 0.05°×0.05°        | 8-day               |
| ERA5-Land | Reanalysis
data     | Skin temperature
Surface pressure
Downward longwave
radiation
Downward shortwave
radiation
2m Temperature
2m Dew point temperature | 0.1°×0.1°          | Monthly             |

 Table 1 Remote sensing and reanalysis datasets used in the RSNP model

| MCD12Q1     |                    |                  |               |        |
|-------------|--------------------|------------------|---------------|--------|
| The water-  |                    |                  |               |        |
| balance-    |                    |                  |               |        |
| based ET on | Reanalysis         | I and cover type | 1 km×1 km     | Appuol |
| dataset of  | data               | Land cover type  | I KIII^I KIII | Annual |
| large river |                    |                  |               |        |
| basins of   |                    |                  |               |        |
| the world   |                    |                  |               |        |
| Version 3   |                    |                  |               |        |
| of the      |                    |                  |               |        |
| Global      | Reanalysis
data | Aridity Index    | 1 km×1 km     |        |
| Aridity     |                    |                  |               |        |
| Index and   |                    |                  |               |        |
| Potential   |                    |                  |               |        |
| Evapotran   |                    |                  |               |        |
| spiration   |                    |                  |               |        |
| Database    |                    |                  |               |        |

**Table 1. Aridity index.**

**Response:** Thank you for your careful review. We have corrected 'arid index' to 'aridity index' throughout the manuscript.

**L93. Was the closure forced on the data. Which method?**

**Response:** Thank you for your comments which helps enhance our manuscript. FLUXNET2015 provide corrected by the energy closure correction factor, and the corrected data is 'LE\_CORR'. To provide further clarification, we have added following statement for clarity:

*"ET observation offered from FLUXNET2015 were corrected by energy balance closure correction factor (Pastorello et al., 2020)."* (Line 175-176)

**Reference:**

Pastorello, G., Trotta, C., Canfora, E., and Papale, D.: The FLUXNET2015 dataset and the ONEFlux processing pipeline for eddy covariance data, Nature Publishing Group, https://doi.org/10.1038/s41597-020-0534-3, 2020.

L100. Some more details on this dataset are needed. A section (2.3) of just few lines is not acceptable. **Response:** Thank you for your valuable feedback. In the revised manuscript, we have significantly expanded the description, provided a more comprehensive explanation of the methodology associated with the WBET dataset, as well as the selection of validation basins. We have expanded the revision as following (Section 2.3.2 in the revision):

"This study applied the water-balance-based ET (WBET) of global typical large river basins dataset to validated global ET datasets at the annual scale (Ma, 2024). This dataset includes WBET data for fifty-six typical large river basins globally from 1983 to 2016, and it is based on measured runoff data from hydrological stations, four precipitation datasets, and three terrestrial water storage datasets. For each river basin under consideration, twelve WBET were generated. Subsequently, the Bayesian three-cornered hat method was employed for weighted averaging to derive the optimal combined WBET dataset. The uncertainty in ET estimation, calculated through error propagation methods, remains within 10% (Ma et al., 2024). This dataset was derived from National Tibetan Plateau Data Center (https://data.tpdc.ac.cn/en/data). We excluded basins that cover less than  $2 \times 10^5$  km2 and ultimately selected 38 basins, and the global distribution of these chosen basins is shown in Fig. 2." (Line 185-195)

Section 2.4. Same as before, some more details are needed. Modelling approach, main inputs, similarity in either inputs or methods, etc.

**Response:** We sincerely appreciate your valuable feedback and apologize for the insufficient description of the datasets, which indeed fell short of proper scientific writing standards. In response to this concern, we have thoroughly expanded the discussion of global ET datasets in the revised manuscript (Section 2.3.3), providing additional details on their modeling approaches and spatial resolutions to enhance clarity. (Line 200-224)

**L116. Nearest neighbour method.**

**Response:** We sincerely thank you for your valuable comments. We have addressed the spelling error throughout the manuscript.

L116. "the differences..." at which time scale? Daily? Monthly? Again, not clear.

**Response:** Thank you for your comments, and we feel sorry for the unclear description. We have revised this sentence which state comparison between RSNP and present ET datasets. The revised statement is as follows:

"This study evaluated the accuracy of global ET datasets regarding the in-situ ET observations at monthly scale, as well as the regional applicability against WBET and the consistency and differences of their spatial and temporal characteristics at the annual scale." (Line 221-223)

**Table 2. This table is not referenced in the text.**

**Response:** Thank you for your comments. Table 2 has now been mentioned at the beginning of Section3.3, where the global ET datasets were first be mentioned. (Line200)

L130. Some more details on the formulations are need. The reader needs to understand the basis of this approach without the need to go reading another full paper. For instance, the first term (Rn-G) is related to the available energy, and it is in common in all ET approach, but what about the other terms? What (Ts4-Ta4) represents? And the logarithmic term with Gs?

**Response:** We sincerely appreciate your suggestions, which have greatly helped to enhance the clarity of our methodology. In response to your valuable comments, we have expanded Section 2.1 to provide a more comprehensive introduction to the methodology (Line 96-126). This includes an explanation of the foundation of the original NP method, the improvements introduced in the SFE-NP method, and how these methods are applied for global estimation. In addition, we also provided Appendix. A and Appendix. B, which detail the derivation of the NP method and the SFE-NP method, and the physical interpretation of each component is elaborated in the supplementary materials.

The meaning of  $\varepsilon \sigma (T_s^4 - T_0^4)$  and  $G_s ln \left(\frac{T_s}{T_0}\right)$  in Eq.1 and Eq.2 can be included in the introduction to the derivation:

When  $T_s$  serves as a generalized coordinate of the system, we can obtain Eq. A4,

$$\frac{\partial HA}{\partial T_s} = \frac{\partial \left(\int_{t_1}^{t_1} \int_A (G_s + H + LE + R_n) dA dt\right)}{\partial T_s} = 0 , \qquad (A4)$$

and obviously we can have  $\partial (G_s + H + LE + R_n)/\partial T_s = 0$ . Among them, the partial derivative of  $R_n$  with respect to  $T_s$  is  $\frac{\partial R_n}{\partial T_s} = -4\varepsilon\sigma T_s^3$ . Under the constraint of a given  $R_n$ , the partial derivative of *LE* with respect to  $T_s$  is  $\partial LE/\partial T_s = 0$  (Wang et al., 2004; Wang et al., 2007) according to the Lagrangian multiplier method. With the energy conservation equation and Fourier's law, further incorporating  $\partial G_s/\partial T_s = G_s/T_s$  (Magyari et al., 1999). Consequently, we can obtain the partial derivation of *H* to  $T_s$  as Eq. A5,

$$\frac{\partial H}{\partial T_s} = 4\varepsilon\sigma T_s^3 - \frac{G_s}{T_s},\tag{A5}$$

and when  $T_s > 0$ ,  $\partial H / \partial T_s$  is evidently a continuous function which can be expressed as Eq. A6 (where  $H_{T_0}$  is the heat flux referenced to surrounding environment when  $T_s = T_0$ ).

$$\int_{T_0}^{T_s} \frac{\partial H}{\partial T_s} dT_s = H_{T_s} - H_{T_0} , \qquad (A6)$$

Integrating from  $T_0$  to  $T_s$ , we can obtain Eq. A7,

$$H_{T_s} = H_{T_0} + \varepsilon \sigma (T_s^4 - T_0^4) - G_s ln\left(\frac{T_s}{T_0}\right),$$
(A7)

among them,  $\varepsilon\sigma(T_s^4 - T_0^4)$  and  $G_s ln\left(\frac{T_s}{T_0}\right)$  is from the derivation and integrating of  $R_n$  and  $G_s$ , respectively.

**Reference:**

- Wang, J., Salvucci, G. D., and Bras, R. L.: An extremum principle of evaporation, Water Resour. Res., 40, https://doi.org/10.1029/2004WR003087, 2004.
- Wang, J., Bras, R. L., Lerdau, M., and Salvucci, G. D.: A maximum hypothesis of transpiration, J. Geophys. Res.: Biogeosci., 112, https://doi.org/10.1029/2006JG000255, 2007.

Eqs. (4) and (5) are not really needed, as they are basic physics. Please expand, instead, on the peculiarity of your method compared to other approaches. How do you "avoid parameters"?

**Response:** We sincerely appreciate your insightful comments. According to your suggestion, we have made following improvements:

(1) We have removed Eqs. (4) and (5) since they are basic equations for relative humidity and saturated water vapor estimation.

(2) We have added supplement to provide the derivation of both the NP and the SFE-NP method (Appendix A and Appendix B) without parametrization of resistances. Traditional methods such as the Penman-Monteith method was builds on the Penman formula by introducing surface resistance to describe the resistance to water vapor transfer between the surface and the near-surface atmosphere, and by using aerodynamic resistance to express the turbulent transfer resistance of heat and moisture in the near-surface atmosphere. In contrast, the reason for NP avoid parameterization

of resistance is that it treats terrestrial ET as a thermodynamic process in a macroscopic state based on Hamilton's principle. Net radiation is regarded as the potential energy of the system, and land surface temperature serves as a generalized coordinate reflecting the energy exchange between the surface and the atmosphere. The dynamics of the system are determined by the Lagrangian L = K - P, the difference between kinetic (K) and potential (P) energies. To a terrestrial ground surface layer, assumed to be isotropic and incompressible without lose of generality, Rn serves as the potential energy. Accordingly, the Lagrangian can be described as:

$$L = \int_{t_1}^{t_2} \int_A U dv dt + \int_{t_1}^{t_2} \int_A G dA dt + \int_{t_1}^{t_2} \int_A (H + LE) dA dt - \int_{t_1}^{t_2} \int_A R_n dA dt,$$
(A1)

where v is volume of the ground surface layer, A is the surface area, and t is the time interval from  $t_1$  to  $t_2$ . The Hamiltonian (HA) HA is described as follows:

$$HA = \int_{t_1}^{t_2} \int_A U dv dt + \int_{t_1}^{t_2} \int_A G dA dt + \int_{t_1}^{t_2} \int_A (H + LE) dA dt + \int_{t_1}^{t_2} \int_A R_n dA dt,$$
(A2)

In the ground surface layer, the heat transfer is traditionally described with Fourier's equation. In the modern theory of general thermodynamics, the equation has been modified to be more general, known as the Maxwell–Cattaneo equation (Tarkenton and Cramer, 1994). On the surface,

 $\int_{t_1}^{t_2} \int_A U dv dt$  vanishes and G shifts to  $G_s$  in each equation. HA is described as follows:

$$HA = \int_{t_1}^{t_2} \int_A G_s dA dt + \int_{t_1}^{t_2} \int_A (H + LE) dA dt + \int_{t_1}^{t_2} \int_A R_n dA dt,$$
(A3)

By virtue of the Hamiltonian system, ground surface temperature  $(T_s)$  serves as a generalized coordinate of the dynamical system in phase space (Strauch, 2009). Based on the Hamiltonian principle, the motion of the system (K + P) remains constant. With respect to  $T_s$ , there has:

$$\frac{\partial HA}{\partial T_s} = \frac{\partial \left(\int_{t_1}^{t_1} \int_A (G_s + H + LE + R_n) dA dt\right)}{\partial T_s} = 0, \tag{A4}$$

And obviously,  $\partial (G_s + H + LE + R_n) / \partial T_s = 0$ .

On the basis of surface energy balance, and assuming that the air temperature is independent on land surface temperature, the partial derivative of  $R_n$  with respect to  $T_s$  is  $\frac{\partial R_n}{\partial T_s} = -4\varepsilon\sigma T_s^3$ , where  $\varepsilon$  is surface emissivity, and  $\sigma$  is the Stephan Boltzmann constant. Under the constraint of a given  $R_n$ ,  $\partial LE/\partial T_s = 0$  (Wang et al., 2004; Wang et al., 2007) according to the Lagrangian multiplier method. With the energy conservation equation and Fourier's law, further incorporating  $\partial G_s/\partial T_s = G_s/T_s$ (Magyari et al., 1999), we obtain the following:

$$\frac{\partial H}{\partial T_s} = 4\varepsilon\sigma T_s^3 - \frac{G_s}{T_s},\tag{A5}$$

When  $T_s > 0$ ,  $\frac{\partial H}{\partial T_s}$  is evidently a continuous function. Mathematically, we obtain the following

equation:

$$\int_{T_0}^{T_s} \frac{\partial H}{\partial T_s} dT_s = H_{T_s} - H_{T_0},\tag{A6}$$

where  $H_{T_s}$  is the sensible heat flux at  $T_s$ , and  $H_{T_0}$  is the heat flux referenced to surrounding

environment when  $T_s = T_0$ . Subsequently, we obtain the following expression:

$$H_{T_s} = H_{T_0} + \varepsilon \sigma (T_s^4 - T_0^4) - G_s ln\left(\frac{T_s}{T_0}\right),$$
(A7)

Coupled with  $H_{T_0}$  is the terrestrial ET in the reference state  $(LE_{T_0})$ . An optional reference state is local thermal equilibrium  $(T_s = T_0)$  for ET. In the original NP method, the conventional equilibrium ET $(LE_E)$  cane be expressed as:

$$LE_E = \frac{\Delta}{\Delta + \gamma} (R_n - G_s), \tag{A8}$$

where  $\Delta$  is the slope of the saturated vapour pressure, and  $\gamma$  is the psychometric constant, was introduced. In practice,  $T_0$  refers to the surface air temperature  $(T_a)$  measured using a standard meteorological instrument. The definition yields  $H_{T_0} = \Delta/(\Delta + \gamma)(R_n - G_s)$  to satisfy the energy conservation correspondingly. Combined with Eq.5, the sensible heat flux can be expressed as:

$$Hs = \frac{\gamma}{\Delta + \gamma} (R_n - G_s) + \varepsilon \sigma (T_s^4 - T_a^4) - G_s ln \left(\frac{T_s}{T_a}\right), \tag{A9}$$

where Ts is the land surface temperature, Ta is the surface air temperature,  $\varepsilon$  is surface emissivity, and  $\sigma$  is the Stephan Boltzmann constant.

Then, the latent heat flux, *LE*, which is equivalent to the actual ET, can be estimated based on the energy budget, and the *LE* estimated by the original NP method is:

$$LE = \frac{\Delta}{\Delta + \gamma} (R_n - G_s) - \varepsilon \sigma (T_s^4 - T_a^4) + G_s ln \left(\frac{T_s}{T_a}\right), \tag{A10}$$

The derived formulas for sensible (Eq.A9) and latent heat fluxes (Eq.A10) contain only observable physical quantities, thus avoiding parameterization of resistance in the non-parametric approaches.

**L156. Why is the resampling needed? Just for a different projection? Not clear.**

**Response:** Thank you for your helpful comments. We are sorry for the mistake in this sentence, because the  $0.05^{\circ}$  GLASS were resampled to  $0.1^{\circ}$  instead of  $0.05^{\circ}$ , and we have revised this error. The revision is as follows:

"And finally, the gap-filled global albedo and BBE, and global aridity index datasets were resampled to 0.1° using the nearest neighbor resampling." (Line 156-157)

**L157. How where water bodies, etc. excluded?**

**Response:** Thank you for your comments. The sentence has been rewritten to indicate that *"water bodies, and permanent ice and snow were excluded according to the classification of MCD12Q1 dataset of each year."*, and we hope the revised expression provides greater clarity and precision. (Line 130)

L158. Here there is the first reference to monthly scale, but it should be made clearer and it should be reported much earlier. Also, is the approach designed for monthly scale? Does eqs. 1-1 and 1-2 valid at monthly temporal scale?

**Response:** We sincerely thank you for your valuable comments about the clarification of the temporal scale of the RSNP ET dataset. The monthly scale should be explicitly stated earlier in the manuscript to avoid ambiguity, and we have made improvements:

a. According to your suggestion, we have introduced the temporal scale in the Abstract: "we improved the Remote Sensed Non-Parametric (RSNP) model based on the NP and SFE-NP method with remote sensing and reanalysis data, and estimated global monthly ET from 2001 to 2019 in the spatial resolution of 0.1°." (Line 21-22)

b. We include our dataset was produced with monthly input images in the revised section '2.2 Framework of Global ET Estimation and Model Fording Data': "GLASS 8-day albedo and BBE data were firstly aggregated into monthly scale,, ..." (Line 153)

L162. Aridity index.

Response: Thank you for your careful review. The incorrect spelling of the word has been corrected.

L163. Aridity. Please fix throughout the text.

**Response:** Thank you for your thorough review, and we appreciate your attention to detail, which helps ensure the quality of our manuscript. The misspelling of "aridity" has been corrected throughout the manuscript.

L163-165. Why 0.65 is used? Please add some reference to support this choice.

**Response:** Thank you for your comments, which helps ensure the quality our manuscript. The United Nations Environment Program (UNEP) defines arid lands based on the aridity index, and when the aridity index is less than 0.65, it is classified as arid area. In the revised manuscript, we have added the references from both the Global Aridity Index dataset and from the UNEP to support this choice. (Line 147)

**Reference:**

Zomer, R. J., Xu, J., and Trabucco, A.: Version 3 of the global aridity index and potential evapotranspiration database, Sci. Data, 9, 409, http://doi.org/10.1038/s41597-022-01493-1, 2022.

UNEP: World Atlas of Desertification - Second Edition, 1997.

Fig. 3. The goal of the upper part of the figure (steps 1 to 5) is not clear. Is this just for gap filling? Very little is said about that in the main text.

**Response:** Thank you for your comments. The step 1 to 5 in original Fig.1 is the gap-filling of the GLASS albedo and BBE, and it is a pre-processing process of model inputs. In the revised Section 2.2, we have restructured the flowchart and rewritten the description of the gap - filling process to present it in a more logical manner:

"The consistency of spatial and temporal resolution in the input images is crucial for ensuring the computability of multi-source remote sensing data. GLASS 8-day albedo and BBE data were firstly aggregated into monthly scale, considering there were still missing pixels of albedo and BBE which would result in missing value of ET result, the nearest day gap-filling method were used to directly fill the missing pixels at the monthly scale." (Line 152-156)

**L168. This sentence is unclear. How was this evaluated?**

**Response:** We sincerely thank you for your comments on this statement. We have looked through the 8-day BBE images with missing values in South Africa, Asia, and Australia, but your comments made us realize that "without reference to" is unreasonable, so we have removed such an inappropriate expression.

L173-176. This part of the pre-processing is very confusing. It needs rewording and expanding. **Response:** Thank you for your valuable suggestion on the statement of pre-processing. We have reorganized the description of the pre-processing in Section 2.2, and we also revised the flowchart according to the description. The revised description of pre-processing is as follows:

"The consistency of spatial and temporal resolution in the input images is crucial for ensuring the computability of multi-source remote sensing data. GLASS 8-day albedo and BBE data were firstly aggregated into monthly scale, considering there were still missing pixels of albedo and BBE which would result in missing value of ET result, the nearest day gap-filling method were used to directly fill the missing pixels at the monthly scale. And finally, the gap-filled global albedo and BBE, and global aridity index datasets were resampled to 0.1° using the nearest neighbor resampling. The land use type was aggregated to 0.1° by the maximum fraction method before adapted into the RSNP model." (Line 152-158)

**Section 3.3 I found this section mostly unnecessary.**

**Response:** We sincerely thank your valuable comments on our manuscript regarding the organization of our manuscript. As you mentioned that the "Validation Method and Accuracy Metrics" is not the main part of this manuscript, and we have removed this section. In addition, the validation process and statistical metrics been contained along with the introduction of validation datasets (Section2.3), and is located at Line161-179:

"The performance of RSNP ET was evaluated through a multi-scale framework. At the site scale, RSNP monthly ET dataset were validated against in-situ EC observations to verify their accuracy against ground observations. While the mismatch between observational footprints and  $0.1^{\circ}$  pixel dimensions could lead to uncertainties in in-situ assessment (Liu et al., 2016), RSNP annual ET were evaluated with water-balance based ET at basins to access the model's effectiveness for the regional scale. Additionally, comprehensive cross-validation was conducted with multiple global ET datasets to examine the consistency and discrepancies in global spatio-temporal patterns. The statistical metrics of assessing the RSNP model including mean bias error (bias), relative error (RE), and Relative Mean Squared Error (RMSE) and Coefficient of determination ( $R^2$ )."

**L182-183. This sentence is not clear.**

**Response:** We sincerely thank you for your valuable feedback on this sentence. We have revised the description of cross-validation process to improve its clarity:

Section 2.3: "Additionally, comprehensive cross-validation was conducted with multiple global ET datasets to examine the consistency and discrepancies in global spatiotemporal patterns." (Line 165-167)

L186. This reference is not needed. These are standard metrics used in validation, not specifically introduced in that research.

**Response:** Thank you for your careful look through of our manuscript and the valuable feedback on this sentence. We have removed the reference of this metrics which is not a specifically point of this research.

L206. Absolute value

**Response:** We sincerely thank you for your careful review. The incorrect spelling of the word has been corrected as 'value'. (L245)

L227-229. This appears to be just 1 point. What is the point to compare this case with all the others? This analysis is very weak.

**Response:** We sincerely appreciate the reviewer's insightful comment regarding the comparison in Fig.7. We acknowledge that drawing conclusions based on a single point lacks statistical significance and weakens the analysis. To address this, we have removed this statement to ensure a more rigorous presentation of our findings. Thank you again for highlighting this important point.

Fig. 8. Differences among models seems mostly systematic, so what is the point of showing multiple years? Wouldn't be better to show the average year? Regional results would also be useful. Global average data are somewhat difficult to analyze.

**Response:** Thank you for your insightful comments on the temporal comparison of global ET datasets, and we have made revisions according to your suggestion. In the original manuscript, the inter-annual variability is presented since the monthly ET values are the direct outputs of the RSNP model, and we compared monthly ET variations to cross-validate the performance of RSNP. According to your comments, we have revised Fig.8 to include ERA5-Land and GLEAM ET dataset for cross-validation, and Fig.8 shows that RSNP exhibits the closest inter-annual variation with GLEAM. In addition, we have also provided the annual variability over the 2003-2018 period in Appendix C (Figure S1) to address the performance of average year, and the Figure S1 reveals that ERA5-Land yields higher magnitudes than RSNP at annual scale, while PEW shows the lowest annual ET. All six global ET dataset showed a pronounced increase from 2009 to 2010. Furthermore, we also presented Africa (Appendix C. Figure S2) annual ET as regional results, for which ET values varied obviously among different ET datasets. RSNP (602.74 mm/yr) presented the highest average ET among Africa, while and other ET datasets ranged from 461.07 mm/yr (PEW) to 562.19 mm/yr (ETMonitor).

Figure S1: Global annual average ET (mm/year) of ETMonitor, PEW, PML-V2, ERA5-Land, GLEAM, and RSNP during 2003-2018.

Figure S2: Global annual average ET (mm/year) of ETMonitor, PEW, PML-V2, ERA5-Land, GLEAM, and RSNP during 2003-2018 over Africa.

L260. What is the difference? Please quantify. This is true for the entire results section, where often qualitative statements such as "is higher" is not accompanied by quantification.

**Response:** We sincerely appreciate your thoughtful comments on our manuscript. We apologize for lacking quantification in our results section, where qualitative statements such as "is higher" are often made without accompanying numerical details. This indeed affects the precision and comparability of our results. We have extended quantification of difference in the highest ET of each dataset, such as in Section3.3.2:

*a.* "In tropical rainforest areas, higher ET values are observed, with RSNP, PML-V2, PEW, and ERA5-Land showing annual ET values in the range of 1300-1500 mm/yr, while ETMonitor and GLEAM display values approximately 1400-1800mm/vr in these regions." (Line 303-305)

*b.* "For instance, the highest annual ET for RSNP, PML-V2, PEW, and ERA5-Land are located in the Malay Archipelago, exceeding 1500 mm/yr. Meanwhile, ETMonitor and GLEAM shows the highest annual ET over 1700mm/yr in both South America and Malay Archipelago." (Line 306-309)

c. "In both the Northern and Southern Hemispheres, the peak ET values of RSNP were observed near the equator (0° latitude), exceeding 1000mm/yr. As latitude increases, the average ET decreases sharply, dropping to nearly 500 mm/yr by around 30° latitude." (Line 319-303) In addition, we also revised other qualitative statements to clarify the details in results and discussions, thank you again for your insightful comments.

L262-265. From the map in Fig. 9, it is not clear this difference in your dataset compared to the other. Also, here and later, there is a lot on emphasis on ET over the desert (missing values, values different than 0, etc.). Is this really that important? Are you expecting notable difference in water budget over these regions.

**Response:** We sincerely appreciate your valuable feedback on this statement. According to your comments, since Fig.9 did not show the difference between RSNP and other datasets, we removed the previous statement, and we have also added Fig.S3 in the supplement to display the differences of annual ET among RSNP and other datasets. Additionally, except the north Africa, we also expanded quantitative comparison for Australia:

"In Australia, PEW, GLEAM, and ERA5-Land showed annual ET lower than 300mm/yr in most areas, while RSNP, ETMonitor, and PML-V2 shows obvious ET variation from 300mm/yr to nearly 700 mm/yr." (Line 316-318)

In addition, ET value plays a crucial role in desert regions, serving as a key component of the water balance. Previous study across 25 basins in Africa revealed that precipitation ranges from ~0 to 300 mm/yr in North Africa's basin, while run-off was ~ 0mm/yr (Karamage et al., 2018). The water balance equation (P=R+E+S, where P is the precipitation, R is the run-off, E is the evapotranspiration, S is the water storage change) would become unbalanced if ET were excluded. Although ET values are typically low in arid regions, they remain an essential consideration in water budget analyses. We have also added related discussion in section 4.1:

"Karamage et al.'s research across Africa basins revealed that both precipitation and run-off varies over basins in North Africa (Karamage et al., 2018). Consequently, missing ET value can compromise the closure of annual water balance assessments." (Line 350-353)

**Reference:**

Karamage, F., Liu, Y., Fan, X., Francis Justine, M., Wu, G., Liu, Y., Zhou, H., and Wang, R.: Spatial relationship between precipitation and runoff in Africa, Hydrology Earth System Sciences Discussions, 2018, 1-27, https://doi.org/10.5194/hess-2018-424, 2018.

**L274. Consistent behaviour with latitude.**

**Response:** Thank you very much for your kind comments. The phrase has been revised as *"consistent behaviour with increasing latitude"* (Line 326)

**L284. Seam?**

**Response:** Thank you for your really helpful suggestion, we apologize for the indeed an inappropriate word usage here. We have replaced "Seam" with "Gap". (Line334)

L295-297. This result, and Fig. 10, raises the question: did you use the same dates for the analyses in Figs. 8 and 9 for all datasets? Average values should be computed on the same samples, so if you used a different number of dates for each dataset (based on availability or coverage) the results will be biased just for that and not for the differences in methodology.

As an example, if one dataset tends to have gaps during cloudy days, its average ET will be higher just because those cloudy days are not included. Please ensure consistency in the results reported. **Response:** We sincerely appreciate your comments regarding spatial and temporal consistency for analyses. We have addressed these valuable concerns from three aspects:

- (1) Fig.8 to Fig.10 is based on the same samples. We collected global monthly ET data of RSNP, ETMonitor, PML-V2, PEW, ERA5-Land, and GLEAM during the period from 2003 to 2018. Fig. 8 collected all available pixel for average monthly ET calculation, and Fig.9's annual ET is the sum of monthly ET. Fig. 10 displays the monthly pixel loss rate of each dataset on a global land surface. We have added statements in the front of Section 4.1 to clarify this issue: "The analysis of temporal and spatial pattern presented in Fig.8 to Fig.10 were from ETMonitor, PEW, PML-V2, ERA5-Land, GLEAM, and RSNP global ET datasets over the 2003-2018 period." (Line284-285)
- (2) For monthly ET estimation, RSNP and other global ET datasets were obtained with clear-sky remote sensing data (albedo and BBE), which is gap-free at the monthly scale. Gaps in other ET datasets primarily from data storage format rather than cloud coverage, as noted in Chaolei Zheng's comments (Comment #3, who published the ETMonitor ET dataset).

(3) While clear-sky ET may slighter higher than all-sky ET, they are conventionally treated as representative of complete monthly values in applied research (Elnashar et.al, 2020; Liu et.al, 2023), as well as the accuracy comparison in similar researches (Ma et.al, 2021; Yu et.al, 2022). Accordingly, the evaluation of ET datasets was performed without all-sky/clear-sky normalization.

Reference:

- Elnashar, A., Wang, L., Wu, B., Zhu, W., and Zeng, H.: Synthesis of global actual evapotranspiration from 1982 to 2019, Earth Syst. Sci. Data, 13, 447-480, http://doi.org/10.5194/essd-13-447-2021, 2021.
- Liu, H., Xin, X., Su, Z., Zeng, Y., Lian, T., Li, L., Yu, S., and Zhang, H.: Intercomparison and evaluation of ten global ET products at site and basin scales, J. Hydrol., 617, 128887, https://doi.org/10.1016/j.jhydrol.2022.128887, 2023.
- Ma, N., Zhang, Y., and Szilagyi, J.: Water-balance-based evapotranspiration for 56 large river basins: A benchmarking dataset for global terrestrial evapotranspiration modeling, J. Hydrol., 630, 130607, http://doi.org/10.1016/j.jhydrol.2024.130607, 2024.
- Yu, L., Qiu, G. Y., Yan, C., Zhao, W., Zou, Z., Ding, J., Qin, L., and Xiong, Y.: A global terrestrial evapotranspiration product based on the three-temperature model with fewer input parameters and no calibration requirement, Earth Syst. Sci. Data Discuss., 2022, 1-33, https://doi.org/10.5194/essd-14-3673-2022, 2022.

L298-299. This statement is confusing. Mu refers to 24% of land surface. Where the 81% comes from? What is a middle-high latitude?

**Response:** Thank you for your valuable feedback. Mu et al. refers to deserts covering 24% of the global land area. The 81% represents the proportion of total global land area located in the Northern Hemisphere, which we derived from statistics. We are sorry that the original statement was unclear. We revised this sentence to better indicate land located northern of 30°N where the gap values mostly located. The revised statement is as follows:

"Barren/deserts regions account for about 24% of the land surface (Mu et al., 2011), and land located north of 30°N latitude accounts approximately 45% of the Earth's total land area." (Line 349-350)

L302. Most of the missing values in the other datasets seems related to desert. How much the water balance can be compromised there when ET is mostly 0 anyway? The missing data is an important point, but it is more relevant in regions when ET is different than 0 when the data are missing. I will focus on these conditions to highlight your point.

**Response:** We sincerely appreciate your insightful comments regarding the discussion of missing values in desert regions. For the statistic of monthly ET, missing values located in both desert regions and above 30°N. The ET value is also important of water balance in arid regions, since previous research over 25 basins in Africa revealed precipitation ranges from ~0 to 300 mm/yr over North Africa's basin, while run-off was ~ 0mm/yr, which addressed the imbalance of water-balance (Karamage et al., 2018). We have also added related discussion to discussion:

"Karamage et al.'s research over Africa basins revealed that both precipitation and run-off varies over basins in North Africa (Karamage et al., 2018). Consequently, missing ET value can compromise the closure of annual water balance assessments." (Line 350-353)

**Reference:**

Karamage, F., Liu, Y., Fan, X., Francis Justine, M., Wu, G., Liu, Y., Zhou, H., and Wang, R.: Spatial relationship between precipitation and runoff in Africa, Hydrology Earth System Sciences Discussions, 2018, 1-27, https://doi.org/10.5194/hess-2018-424, 2018.

**L305. Shifts.**

**Response:** We thank you for catching this typo. The excessive full stop has been removed in the revised manuscript. (Line 357)

**Fig. 11. Monthly availability... How is this monthly? Not clear.**

**Response:** We sincerely appreciate your comment on Fig. 11, and thank you for pointing out the lack of clarity regarding the monthly representation. We have revised the title of Fig.11 as '*Figure 12: Spatial distribution of monthly ET pixels available ratio during 2003-2018 for global ET datasets (water, and permanent ice and snow were excluded).*' to clarify the time-span for analysis. (Line 361-362)

L317. This dataset is seamless because is not a remote sensing-based product. If you try to use skin LST from satellite, then you would have a RS dataset but with some gaps.

**Response:** We sincerely appreciate your valuable comments, which significantly helps to enhance the academic quality of our paper. To address your valuable comments, we have removed emphasis on "our remote sensing ET data" since we used both remote sensing and reanalysis data as model forcing. The revised statement refers to *"The RSNP model enables gap-free global ET estimation without reliance on calibration or parameterization."* (Line 374-375). Furthermore, the generation of long temporal scale (monthly/annual) LST composites involves aggregation of short temporal scale (instantaneous/daily) observations, with gap-filling applied during monthly compositing. Consequently, missing values in the model output ET datasets are not only attributed to LST data. Considering that recent studies have published high accuracy global gap-free LST datasets (Jia et al., 2023; Hong et al., 2021; Liu et al., 2023), we will further adopt remote sensing LST product to update the RSNP ET dataset, providing a more detailed estimation at a higher spatial resolution.

**Reference:**

- Hong, F., Zhan, W., Göttsche, F.-M., Lai, J., Liu, Z., Hu, L., Fu, P., Huang, F., Li, J., and Li, H.: A simple yet robust framework to estimate accurate daily mean land surface temperature from thermal observations of tandem polar orbiters, Remote Sens. Environ., 264, 112612, https://doi.org/10.1016/j.rse.2021.112612, 2021.
- Jia, A., Liang, S., Wang, D., Ma, L., Wang, Z., and Xu, S.: Global hourly, 5 km, all-sky land surface temperature data from 2011 to 2021 based on integrating geostationary and polar-orbiting satellite data, Earth Syst. Sci. Data, 15, 869-895, https://doi.org/10.5194/essd-15-869-2023, 2023.

Liu, X., Li, Z.-L., Li, J.-H., Leng, P., Liu, M., and Gao, M.: Temporal upscaling of MODIS 1-km instantaneous land surface temperature to monthly mean value: Method evaluation and product generation, IEEE Transactions on Geoscience Remote Sensing, 61, 1-14, http://doi.org/10.1109/TGRS.2023.3247428, 2023.

L320. This statement is very confusing to me, first because skin LST from ERA5-land relies on these resistances, and second because the methodology does not explain how the method get rid of the resistances.

**Response:** We sincerely appreciate your insightful comment regarding how our method eliminates resistances. We apologize for any confusion caused by the unclear statements in the original text. To address this, we will clarify the issue from the following aspects:

(1) We apologize for the incorrect description of this statement, and we have revised this statement to emphasize that the model is based on methods which avoid parameterization of resistances, and the revised statement is "Conversely, the RSNP based on Hamiltonian principle and remains a diagnostic model independent from empirical resistance or calibration and is helpful for eliminating uncertainties in global terrestrial water-energy budget researches.". (Line 396-399)

(2) To clarify the process of avoiding parameterization, we have added supplement to provide the derivation of both the NP and the SFE-NP method (Appendix A and Appendix B). Traditional methods such as the Penman-Monteith method was builds on the Penman formula by introducing surface resistance to describe the resistance to water vapor transfer between the surface and the near-surface atmosphere, and by using aerodynamic resistance to express the turbulent transfer resistance of heat and moisture in the near-surface atmosphere. In contrast, the reason for NP avoid parameterization of resistance is that it treats terrestrial ET as a thermodynamic process in a macroscopic state based on Hamilton's principle. Net radiation is regarded as the potential energy of the system, and land surface temperature serves as a generalized coordinate reflecting the energy exchange between the surface and the atmosphere. The derived formulas for sensible and latent heat fluxes only contain observable physical quantities. In the RSNP model, the model inputs include surface net radiation, soil heat flux, air temperature, land surface temperature, and relative humidity, they can directly be derived from remote sensing and reanalysis dataset.

(3) We sincerely appreciate your insightful comment regarding the potential uncertainties in the estimation accuracy of ERA5-Land skin LST due to its reliance on the parameterized H-TESSEL model. To address this, we would incorporate global gap-free remote sensing LST products (e.g., those developed by Hong et al., Jia et al., and Liu et al.) as model forcing in future improvements of the RSNP model. Furthermore, we have acknowledged the limitations of the LST input in the discussions as "Additionally, utilizing gap-free global LST products as model inputs offers stronger physical foundations and reduced parametric dependence relative to reanalysis datasets, which would better controlling uncertainty at the data input level." (Line 403-407). We hope these revisions could help to improve our dataset in the future.

**L327. Our dataset...**

**Response:** Thank you for pointing out the spelling error of the original manuscript. The statement have been improved in the revised manuscript, and we have checked and corrected spelling error throughout the whole manuscript. Thank you again for your attention to detail, which has helped improve the quality of our paper.

L328-332. This sentence is very confusing. Please reword.

**Response:** Thank you for your insightful comment regarding the clarity of our statements. We appreciate your feedback, and we have addressed the confusion in the sentence you pointed out. The revise sentence is as following:

"Moreover, the RSNP ET dataset, with a spatial resolution of 0.1°, provides more detailed ET information compared to PML-V2 (0.5 km), PEW (0.1°), and GLEAM (0.25°). This suggests that spatial resolution alone does not determine the accuracy of ET datasets (Stisen et al., 2008; Zheng et al., 2022)." (Line 385-388)

L333. ERA5-land has already an ET product. You should include in your analysis a comparison with that product, as it is based on mostly the same forcings and is produced together with the skin LST used in this study. What is the added value of your methodology compared with what is already there?

Response: Thank you for your valuable suggestions which are important for our manuscript.

In response to your suggestion regarding the necessity of the RSNP model's dependence on resistance parameters, we have included ERA-5Land ET dataset for cross-validation in the revised manuscript. Consequently, the results from cross-validation revealed that, RSNP performed better than ERA5-Land under vegetated and saturated land covers, "RSNP shows higher in-situ accuracy than ERA5-Land for vegetated and saturated land covers, and the accuracy improvement at the wetland sites is significant, with RMSE reducing from 65.21 mm/month to 20.60 mm/month." (Line 252). In addition, RSNP demonstrates superior performance in capturing spatial details of ET patterns, even ERA5-Land and RSNP share most of the model forcing at the same spatial resolution (0.1°). Corresponding comparison and discussion have been added to Section4.2, such as: "For instance, in South America, ERA5-Land shows minimal spatial differentiation in ET between low and high vegetation areas, while RSNP successfully captures the gradual transition of ET variation, better reflecting the actual surface heterogeneity." (Line 383-385). In conclusion, the added value of proposed RSNP model existed in both improve accuracy and reduce uncertainties in ET estimation. We have attached these importances in the discussion and we hope these revisions would improve our manuscript. Thank you again for these insightful suggestions.

---

## Author Comment (AC2)

**Response to Anonymous Referee #2:**

Dear reviewer, we sincerely appreciate your time and effort in reviewing our manuscript and providing valuable feedback to help improve our work. In the reply, the reviewer's comments are in black, our responses are in blue, and quotes from the revised manuscript are in *orange italics*.
* * *
The manuscript provides a new global ET dataset, which is meaningful for the detection of long-term global ET variation and water resource detection. Then that dataset is validated by FLUXNET sites and water-balance ET data. The subject of the manuscript is within the scope of ESSD. It is worth publishing in the journal provided a major revision following the comments given as below:

The concept of "Nonparametric Approach" which was mentioned in the title, it suggested to briefly explain the core concept of this method in the Abstract.

**Response:** We sincerely appreciate your valuable suggestion regarding the explanation of the nonparametric approach in the Abstract. To address this problem, we have revised the text accordingly. The updated statement is as follows:

"*The nonparametric (NP) method and Surface Flux Equilibrium-nonparametric (SFE-NP) method are ET estimation approaches that without the parametric of resistance parameters.*" (Line20-21)

L 28: "our dataset offers a continuous and seamless ET dataset suitable for global research." The repeated use of the dataset, and the sentence should be simplified.

**Response:** We sincerely appreciate your feedback on this statement. As suggested, we have revised the sentence to improve clarity and precision. The revised statement is as follows:

"*Furthermore, RSNP provides continuous and gap-free global ET.*" (Line27)

L 28: "This study contributes to the advancement of global ET estimation and informs future water balance studies." It is too general and lacks specific descriptions of your contributions.

**Response:** We sincerely appreciate your valuable suggestion regarding the need for a more specific description of our study's contributions. As suggested, we have revised the concluding sentence of the Abstract to better highlight the key advancements of our work. The updated text now clearly states:

"*This study advances global ET estimation by eliminating the need for resistance parameterization, and the RSNP ET dataset directly supports improved water resource management and climate modeling efforts.*" (Line28-30)

L 37: Please extend the description with how and why the distribution of these flux sites across the global land surface influenced the accuracy of estimating global ET.

**Response:** Thank you for your valuable suggestion, which has helped improve the clarity of our manuscript. According to your comments, we have refined our description to more explicitly highlight the discontinuous temporal coverage and discrete spatial distribution of ground-based observations limit their ability to estimate global ET. And the revised sentence lays the groundwork for subsequently discussing the advantages of using remote sensing to estimate regional ET. The updated text now reads:

"*This discontinuous and uneven sampling makes point-scale observations particularly inadequate for capturing the spatiotemporal dynamics of regional water and energy cycles. Notably, the underlying surface within a region often demonstrates greater homogeneity compared to the localized conditions represented by individual stations, further emphasizing the limitations of relying solely on discrete point measurements.*" (Line39-43)

L 39: The word "conduct" is repeatedly used, please rewrite the sentence.

**Response:** We sincerely appreciate your valuable suggestion regarding the repeated use of the word "conduct." As per your comment, we have revised the sentence to improve clarity and avoid redundancy, and the revision is as following:

"*The ability to perform periodic and repetitive observations over regions, coupled with its cost-effectiveness, enables remote sensing conducting global ET observation (Liu et al., 2022; Zhang et al., 2016).*" (Line43-45)

L 64: Although the last paragraph of Introduction mentions the non-parametric method (NP) and the surface flux equilibrium-non-parametric method (SFE-NP), there is no detailed explanation of the principles of these methods and their advantages over the traditional parametric methods. Nor is it explained how these methods avoid the complex parametric process and how they improve the accuracy and applicability of ET estimation.

**Response:** We sincerely appreciate your valuable comments regarding the need for more detailed explanations of the NP and SFE-NP methods. In response to your suggestions, we have made the following revisions to the manuscript:

(1) In the last paragraph of the Introduction section, we have added a more comprehensive explanation of both methods' principles and advantages:

"*Evaporation is the phase change process where water molecules transition from liquid to vapor, and thermal driving is the primary mechanism governing terrestrial evaporation. Hamilton's principle offers a physical insight into the macro-state processes to mechanics and describe thermodynamics.*" (Line74-76)

"*The original nonparametric (NP) method is based on the Hamiltonian principle that terrestrial ET follows in the macroscopic state, with surface temperature as a generalized coordinate of the Hamiltonian, and combining with the equilibrium ET (Liu et al., 2012), the original NP method is in a simple analytical form without parameterization of aerodynamic resistances.*" (Line76-80)

(2) As supplementary material, we are providing a detailed derivation of the NP and SFE-NP formulations (Appendix A and B), which includes the mathematical foundations of both methods and step-by-step derivations of key equations.

L72: In the Introduction section, although the RSNP model was mentioned, but there is no detailed explanation of how this model solves the problems of existing models and its unique contribution in global ET estimation. The research goals should be more specific and clearer.

**Response:** Thank you very much for your comments regarding the gap that this study addresses. As you suggested, a clearer explanation for RSNP's novelty and specific research goal is of vital importance in the Introduction section. After introducing the NP and SFE-NP methods, we pointed out the novelty of developing a global dataset based on the NP approaches, and the crucial role

that RSNP will play in reducing systematic errors in the global land surface ET analysis of multi-dataset integration. The revised statement of research goal is as follows:

*"Consequently, developing a global ET dataset based on NP approaches helps to reduce uncertainty by eliminating the reliance on resistance parameterizations. A globally improved model based on NP approaches (namely RSNP model) is proposed in this paper, from which a global, gap-free ET dataset has been produced. As a novel Hamiltonian principle-based global ET model, RSNP model would especially reduce the systematic errors of the datasets based on the same principle (e.g. the PM method) when integrating different global ET datasets for global change research."* (Line88-94)

L84: Please explain how the 1 km resolution of MODIS land cover data was reconciled with the 0.1° resolution of other datasets. Was any downscaling or upscaling applied, and if so, what methods were used?

**Response:** We sincerely thank you for your constructive feedback, which helps to improve the clarity of our manuscript. In this research, the 1km MODIS land cover classification dataset was resampled with the maximum fraction method. According to your comments, we added the following description in Section2.2:

*"The land use type was aggregated to 0.1° by the maximum fraction method before adapted into the RSNP model."* (Line157-158)

L109: The RSNP model's input data are mainly from ERA5-Land, and ERA5-Land also provides a data set of actual ET. However, the section of cross-validation of RSNP does not reflect the comparison with ERA5-Land.

**Response:** We sincerely appreciate your insightful comment regarding the comparison between RSNP and ERA5-Land ET datasets. Since the radiation variables and surface temperature in RSNP model are derived from ERA5-Land, we acknowledged that a systematic comparison would better highlight the unique characteristics and advantages of the proposed RSNP. In response to your suggestion, we have added the following analysis in the revised manuscript:

(1) We added the ERA5-Land to the validation and comparison:

The revise scatter plot of model validation shows that RSNP has a more concentrated scatter density distribution than ERA5-Land, especially less underestimations (Fig.4). For different land covers, RSNP shows higher in situ accuracy than ERA5-Land for vegetated land covers, and the accuracy improvement at the wetland sites is significant, with RMSE reducing from 65.21 mm/month to 20.6 mm/month (Fig.5). At the basin scale, statistical comparisons reveal that RSNP's RMSE, bias and R² values fall within similar ranges as other global ET products, suggesting equivalent capability in capturing ET dynamics at the basin scale.

[Figure]

**Figure 4: Comparison of estimated ET and observed ET over FLUXNET2015 sites. The relative mean square error (RMSE) and the bias are both in mm/month.**

[Figure]

**Figure 5: Comparison of estimated ET and observed ET over FLUXNET2015 sites at ten types of land covers, including MF (Mixed Forest), GRA (Grassland), SAV (Savanna), WSA (Woody Savanna), EBF (Evergreen Broadleaf Forest), CRO (Cropland), DBF (Deciduous Broadleaf Forest), ENF (Evergreen Needleleaf Forest), WET (Wetland), OSH (Open Shrublands). The relative mean square error (RMSE) and bias are both in mm/month.**

(2) We added the ERA5-Land ET dataset to the spatial distribution analysis:

Although RSNP and ERA5-Land ET sharing identical input parameters, they fundamentally different in algorithmic principles - physical parameterization versus Hamiltonian-based nonparametric formulation, which lead to distinct spatial patterns in estimated terrestrial ET. From the comparison in regional areas, RSNP shows more details than ERA5-Land such as in the South Africa (Fig.12).

*"For instance, in South America, ERA5-Land shows minimal spatial differentiation in ET between low and high vegetation areas, whereas RSNP successfully captures the gradual variation of ET, better reflecting the actual surface heterogeneity."* (Line383-385)

[Figure]

**Figure 13: Spatial pattern of global ET datasets in typical regions in 2014. Columns from left to right: (a) Northwest China; (b) Northwest America; (c) Southwest America.**

L110: Several acronyms (e.g., PT-JPL) are introduced without full definitions upon first mention, which hinders readability for non-specialist audiences. Ensure all abbreviations are spelled out at first occurrence.

**Response:** We sincerely appreciate your careful reading and constructive suggestion regarding acronym definitions. In response to this comment, we have spelled out all acronym when they firstly occurred at the Introduction section of the revised manuscript, such as:

*"PEW is constructed based on a unified water balance model derived from the generalized proportionality hypothesis and incorporating available water control into the Priestly Taylor-Jet Propulsion Laboratory (PT-JPL) algorithm, ..."* (Line214-216)

L138: There is an error in Equation2 for calculating net surface radiation, and it should be revised.

**Response:** We greatly appreciate your careful examination and valuable correction, and we sincerely apologize for the error in Eq.2 regarding the calculation of net surface radiation. We have added the 4th power superscript to the final term of the net surface radiation equation as follows: (Line124)

$$R_n = (1 - \alpha)R_{sd} + R_{ld} - \varepsilon_s \sigma T_s^4, \tag{3}$$

L180: "Direct validation is composed of validation at the point scale and validation at the basin scale". It is necessary to elaborate on the specific differences and complementarities of these two validation methods, and to verify the validity and reliability of the model from which aspects respectively?

**Response:** We sincerely appreciate your insightful suggestion regarding the clarification of our validation approaches. We have substantially revised the statement to better explain the differences and complementarities between point-scale and basin-scale validation methods, as well as their respective roles in verifying model performance. The revised statement is located in Section2.3:

*"The performance of RSNP ET was evaluated through a multi-scale framework. At the point scale, RSNP monthly ET were compared against in-situ Eddy Covariance observations to verify their accuracy against ground observations. For the regional scale, RSNP annual ET were evaluated with water-balance based ET at basins to access the model's effectiveness. Additionally, comprehensive cross-validation was conducted with multiple global ET products to examine the consistency and discrepancies in global spatio-temporal patterns."* (Line161-167)

L196: Figure 4 reflects the scatter density with stretched colors, but a color band indicating whether red or blue represents a high or low density is missing?

**Response:** We sincerely appreciate the reviewer's careful observation regarding the clarity of Figure 4. In response to this comment, we have revised the figure to include a labeled color bar indicating the density scale, where red represents high-density regions with frequent data points and blue represents low-density regions with sparse data points), and the revised Fig.4 is as follows:

[Figure]

**Figure 4: Comparison of estimated ET and observed ET over FLUXNET2015 sites. The relative mean square error (RMSE) and the bias are both in mm/month.**

L228: "RSNP has certain advantages in monitoring basin or regional ET on a global scale", but it does not specify what these advantages are. Similar general statements in the article should be thoroughly proven and expanded.

**Response:** Thank you very much for your feedback regarding the need for more specific and evidence-based statements about RSNP's advantages. In response to this comment, we have thoroughly revised the statements to explicitly highlight RSNP's strengths.

(1) We have added the statement to emphasis RSNP's ability of estimating ET at basin scale: *"Comparative analysis revealed that RSNP consistently outperformed ETMonitor, PEW, and GLEAM at the basin scale, exhibiting both lower RMSE and slightly higher R² across all validation basins. Nevertheless, although most ET datasets exhibited substantial biases in basins with annual ET exceeding 1200 mm/yr, RSNP demonstrated the least overestimation among them."* (Line265-269)

(2) We have concluded the Hamiltonian principle-based ET dataset is qualified for regional analysis: *"The consistency between RSNP and other global ET datasets in WBET validation confirms the reliability of the Hamiltonian principle-based and resistance parameterization-free model for global land surface ET estimation, providing strong evidence to support its application and adoption."* (Line272-275)

L330: The expression "unsatisfactory performance" is not specific enough. It is suggested to change it to "limited accuracy".

**Response:** We sincerely appreciate your constructive suggestion to improve the precision of our manuscript. Following your recommendation, we have carefully revised the expression to better reflect the meaning, and the revised sentence is as follows:

*"This highlights that the performance of ET datasets is not solely determined by spatial resolution.*

*(Stisen et al., 2008; Zheng et al., 2022)."* (Line 387)

---

## Author Comment (AC3)

**Response to Thomas Van Niel:**

Dear reviewer, we sincerely appreciate your time and effort in reviewing our manuscript and providing valuable feedback to help improve our work. In the reply, the reviewer's comments are in black, our responses are in blue, and quotes from the revised manuscript are in *orange italics*.

**Overview**

The study introduces a global terrestrial evapotranspiration (ET) dataset (2001–2019, 0.1° resolution) using the Remote Sensed Non-Parametric (RSNP) model, which avoids complex parameterization by leveraging nonparametric (NP) and Surface Flux Equilibrium-Nonparametric (SFE-NP) approaches with remote sensing and reanalysis data. Validation against FLUXNET and water-balance ET showed comparable accuracy to existing datasets (ETMonitor, PML\_V2, PEW). RSNP offered more complete global coverage by reducing missing values, especially in arid regions. I personally learned a great deal from my own research into the Hamiltonian approach used and came away inspired to test new ideas. However, almost none of this understanding came directly from the paper, which largely glosses over, arguably, the most compelling reason to publish the work. Because of the novelty of the approach used to generate the dataset, I would very much like to see this paper published. However, it would require a substantial effort to make it ready for publication, in my opinion. I describe 5 major comments/concerns that I have about the manuscript in its current state. These should be explicitly addressed in the author's response. The intent of my comments is only to help improve the manuscript. I then provide a list of minor issues.

**Major Comments/Concerns:**

1.) Insufficient Explanation of the Hamiltonian Approach: The manuscript does not provide sufficient detail on the Hamiltonian microstate approach, making it unclear why this method was chosen over a standard deterministic model. The novelty of this approach is underemphasized, despite it representing a fundamental departure from traditional surface energy balance (SEB) modelling. The lack of explanation makes it difficult for readers to fully understand the rationale behind this choice and assess its advantages. I only realized the significance of the approach after questioning the formulation of Eq. (1-1) and (1-2) and conducting my own research. The authors should provide a much clearer and more detailed explanation of the Hamiltonian (variational) method and explicitly highlight how it differs from conventional deterministic SEB modelling approaches. Strengthening this discussion would better justify its use and emphasize the novelty of the study.

**Response:** We sincerely appreciate your suggestion regarding the Hamiltonian approach. In response, we have substantially expanded the Introduction and Section2.1 to provide a more detailed explanation of the assumption of Hamiltonian's principle and theoretical foundations of the original NP methods, the limitations in the original NP method, as well as the improvement of the SFE-NP method. In the revised manuscript, we also upload supplements (Appendix A and B) to include the derivation of each parameter followed the Hamilton's principle, and we hope this revision can enhance the clarity of independent of resistance parameterization.

**a. Introduction:**

"Evaporation is the phase change process where water molecules transition from liquid to vapor, and thermal driving is the primary mechanism governing terrestrial evaporation. Hamilton' s principle offers a physical insight into the macro-state processes to mechanics and describe thermodynamics. The original nonparametric (NP) method is based on the Hamiltonian principle that terrestrial ET follows in the macroscopic state, with surface temperature as a generalized coordinate of the Hamiltonian, and combining with the equilibrium ET (Liu et al., 2012), the original NP method is in a simple analytical form without parameterization of aerodynamic resistances. To address NP method's applicability in arid areas, the surface flux equilibrium (SFE) with relative humidity was introduced to develop the SFE-NP method (Pan et al., 2024)." (Line 74-82)

**b. Section2.1:**

"According to Hamilton's principle, net radiation ( $R_n$ ) represents the potential energy in a macro-state system, while soil heat flux ( $G_s$ ), latent heat flux (LE), and sensible heat flux ( $H_s$ ) collectively constitute the kinetic energy. In a macro-state system, terrestrial ET can be treated as a mechanical and thermodynamic process following Hamilton's principle, and temperature ( $T_s$ ) is an intensive thermodynamic indicator of a macro-state system. By adopting  $T_s$  as generalized coordinate within the Hamiltonian system and incorporating equilibrium evaporation, the original NP method expresses LE as a function of  $R_n$ ,  $G_s$ ,  $T_s$  and air temperature ( $T_a$ ), which eliminates the need for parameterizing resistance terms (Liu et al., 2012)" (Line 97-104)

"The application of the NP method in remote sensing retrieval has shown high accuracy in humid regions, however, its effectiveness appears to be limited in arid regions (Hsieh et al., 2022; Yang et al., 2016). The primary limitation is that the conventional equilibrium ET employed in the original NP method is limited to the wet situation and loss the applicability to arid region. To address this limitation, the SFE-NP method has been introduced by replacing the conventional equilibrium ET with an equilibrium ET estimation based on the relative humidity (RH) budget, aiming to enhance ET estimation from unsaturated surfaces (Pan et al., 2024)." (Line 108-114)

**2.) Further thinking/justification of the surface partitioning constraint:**

The function, ln(Ts/Ta), that is in both Eq. (1-1) and (1-2) would seem to me to be very insensitive to change within a realistic range of naturally occurring terrestrial land surface and air temperatures. I feel like it would, thus, fail to effectively scale energy partitioning. For example, when I calculate the output for temperatures in Kelvin for a few realistic terrestrial temperature examples, I get:

- Ts = 308.15 K (35°C), Ta = 298.15 K (25°C)  $\rightarrow \ln(Ts/Ta) \approx 0.033$
- Ts = 288.15 K (15°C), Ta = 298.15 K (25°C)  $\rightarrow \ln(Ts/Ta) \approx -0.034$

As can be seen from the two examples above, the function's output is very small. The reason for this is that if the temperatures are in Kelvin, then the difference between Ts and Ta is relatively very small compared to either of Ts or Ta, resulting in values only negligibly different from unity. When the ln is taken of values near one, they are always small. This would, subsequently make it behave almost linearly and prevent the function from capturing the expected nonlinear shift in energy partitioning from latent heat to sensible heat as the surface dries. Additionally, if temperatures are expressed in degrees Celsius, the function becomes physically invalid, as it involves taking the logarithm of a ratio that can include negative values or be divided by zero. These issues suggest to me that ln(Ts/Ta) is not a suitable scaling function for partitioning surface energy fluxes within the Hamiltonian framework. Apologies if I've got this wrong. I'd appreciate to hear from the authors specifically if I've made a mistake in my interpretation. If I am right, then at the very best, this function is doing almost nothing to partition the sensible and latent heat fluxes. The authors may

be better off looking into a more appropriate function of Ts and Ta, which might improve the partitioning of latent and sensible heat fluxes. If this function has nearly no impact on the model, then what does it say about the reason the model outputs very reasonable ET estimates? Is it because the ERA5 data are doing most of the work? I discuss this more below.

**Response:** Thank you very much for your feedback on the NP method. Our response according to these comments contains three aspects:

(1) The NP method was derived based on both the Hamilton's principle and the conventional equilibrium ET. When  $T_s$  serves as a generalized coordinate of the system, we can obtain Eq. A4, and obviously we can have  $\partial(G_s + H + LE + R_n)/\partial T_s = 0$ . Among them, the partial derivative of  $R_n$  with respect to  $T_s$  is  $\frac{\partial R_n}{\partial T_s} = -4\varepsilon\sigma T_s^3$ . The partial derivative of *LE* with respect to  $T_s$  is  $\frac{\partial R_n}{\partial T_s} = -4\varepsilon\sigma T_s^3$ . The partial derivative of *LE* with respect to  $T_s$  is  $\partial LE/\partial T_s = 0$  (Wang et al., 2004; Wang et al., 2007) . According to the Lagrangian multiplier method, and the energy conservation equation and Fourier's law, further incorporating  $\partial G_s/\partial T_s = G_s/T_s$  (Magyari et al., 1999). Consequently, we can obtain the partial derivation of *H* to  $T_s$  as Eq. A5, and when  $T_s > 0$ ,  $\partial H/\partial T_s$  is evidently a continuous function which can be expressed as Eq. A6 (where  $H_{T_0}$  is the heat flux referenced to surrounding environment when  $T_s = T_0$ ). Integrating from  $T_0$  to  $T_s$ , we can obtain Eq. A7. Consequently, the logarithmic term  $ln(T_s/T_a)$  is not chosen as a scaling function, but a term from the derivation and integrating of  $G_s$ . And details of derivation are included in the supplement (Appendix A) and research papers of the original NP and SFE-NP

$$\frac{\partial HA}{\partial T_s} = \frac{\partial \left(\int_{t_1}^{t_1} \int_A (G_s + H + LE + R_n) dA dt\right)}{\partial T_s} = 0, \tag{A4}$$

$$\frac{\partial H}{\partial T_s} = 4\varepsilon\sigma T_s^3 - \frac{G_s}{T_s},\tag{A5}$$

$$\int_{T_0}^{T_s} \frac{\partial H}{\partial T_s} dT_s = H_{T_s} - H_{T_0},\tag{A6}$$

$$H_{T_s} = H_{T_0} + \varepsilon \sigma (T_s^4 - T_0^4) - G_s ln\left(\frac{T_s}{T_0}\right),$$
(A7)

(2) The land surface temperature and air temperature are in Kelvin in this paper. In addition, Hsieh had conducted the validation of the three terms in NP method (Hsieh et al., 2022), and indicated that the first term  $\frac{\Delta}{\Delta + \gamma} (R_n - G_s)$  is the major component contributes to LE, the second term  $\varepsilon_s \sigma (T_s^4 - T_a^4)$  accounts about  $\pm 10\%$ , and the third term  $G_s ln (\frac{T_s}{T_0})$  is close to zero. Therefore, no matter the unit of temperature is K or C°, the Ts/T0 affected the ET insignificantly. In addition, prior to the development of the proposed RSNP model, which utilizes GLASS and ERA5-Land data as inputs, the NP method had already been successfully applied with Moderate Resolution Imaging Spectroradiometer (MODIS) and China Meteorological Administration Land Data Assimilation System (CLDAS) in the Lower Mekong River basin and the Poyang Lake basins. The selection of ERA5-Land data for current study was driven by its global coverage and comprehensive inclusion of relevant parameters.

(3) Since the first term of NP method accounts for the majority of the LE results, the improvements to the non-parametric approaches based on Hamiltonian principles and equilibrium evapotranspiration primarily focus on refining the first term. For instance, the SFE-NP approach

introduces an equilibrium state based on relative humidity to enhance its applicability over nonsaturated underlying surfaces (Pan et.al, 2024). We will also continue to explore the improvement of the accuracy of NP method in the future.

Reference:

- Hsieh, C.-I., Chiu, C.-J., Huang, I.-H., and Kiely, G.: Estimation of Latent Heat Flux Using a Non-Parametric Method, Water, 14, 3474, http://doi.org/10.3390/w14213474, 2022.
- Pan, X., Yang, Z., Liu, Y., Yuan, J., Wang, Z., Liu, S., and Yang, Y.: A non-parametric method combined with surface flux equilibrium for estimating terrestrial evapotranspiration: Validation at eddy covariance sites, J. Hydrol., 631, 130682, http://doi.org/10.1016/j.jhydrol.2024.130682, 2024.

3.) Justification for a New Global ET Model:

One of the key questions that arises is whether a new global ET model is truly needed, particularly given that the proposed dataset appears to perform similarly to existing models. The primary stated advantage of the dataset is that it is gap-free, but this claim is not inherently compelling, as the seamless nature of the data appears to be a result of gap-filling through averaging rather than a fundamentally new methodological breakthrough. The authors should clarify what specific advancements their approach offers beyond convenience, particularly in relation to existing global ET datasets. After a very quick web search I found several existing global datasets, see below. The list of datasets in not intended to be comprehensive. The authors should, in my opinion, include a more comprehensive summary of the current global ET datasets and then justify the need for a new one. A table that summarises the available datasets and classifies them into groups by some relevant criteria would be very helpful.

- Global land surface evapotranspiration monitoring by ETMonitor model driven by multisource satellite earth observations https://www.sciencedirect.com/science/article/pii/S0022169422010149
- A global dataset of terrestrial evapotranspiration and soil moisture dynamics from 1982 to 2020 https://www.nature.com/articles/s41597-024-03271-7
- On the divergence of potential and actual evapotranspiration trends: An assessment across alternate global datasets https://doi.org/10.1002/2016EF000499
- A global terrestrial evapotranspiration product based on the three-temperature model with fewer input parameters and no calibration requirement Earth Syst. Sci. Data, 14, 3673–3693, 2022 https://doi.org/10.5194/essd-14-3673-2022
- A Comprehensive Evaluation of Five Evapotranspiration Datasets Based on Ground and GRACE Satellite Observations: Implications for Improvement of Evapotranspiration Retrieval Algorithm https://www.mdpi.com/2072-4292/13/12/2414
- Multi-scale evaluation of global evapotranspiration products derived from remote sensing images: Accuracy and uncertainty https://www.sciencedirect.com/science/article/pii/S0022169422005571
- Global Evapotranspiration Datasets Assessment Using Water Balance in South America https://www.mdpi.com/2072-4292/14/11/2526

GLEAM4 https://repository.kaust.edu.sa/items/0980d173-e356-48b9-9bae-19c81d830eb7

Response: We sincerely appreciate your thoughtful comments regarding the justification for our

new global ET model. We have made below revisions to highlight the importance of our study regarding your valuable suggestions:

(1) We have carefully revised the manuscript to better highlight the necessity of establish the RSNP model, particularly its foundation in Hamiltonian principles that provide clear physical meaning to the derived ET estimates, which is a key distinction from existing datasets. Regarding the need for a new dataset, while many studies rely on multi-dataset integration, most input global ET datasets share similar basic methodology (e.g., PM method, PT method, and surface energy balance residual method) that may introduce correlated systematic errors. Our physics-based RSNP global ET dataset offers an independent alternative that could help mitigate such issues in future synthesis studies. The relative statements are as follows:

"By evaluating 25 global ET datasets with site observations and their spatial patterns. Tang et.al refer that ET dataset produced based on similar algorithms tend to have high consistency in annual magnitude and spatial distribution (Tang et al., 2024). Therefore, developing a global ET dataset based on well-defined physical mechanisms remains a critical need in ET research. Moreover, integrating datasets with reliable accuracy and clear physical significance can enhance the robustness of analytical results in global data synthesis." (Line 68-73)

(2) We have cited researches which have already compared existing global ET products, and theses reference could provide a much comprehensive intercomparison of global ET datasets (Zheng et al., 2019; Cheng et al., 2020; Elnashar et al., 2021; Ma and Zhang, 2022; Liu et al., 2023; Tang et al., 2024) (Line54-59). In addition, we have also added a table to summarize typical published global ET datasets (Appendix C. Table 1), including eight remote sensing datasets, two reanalysis datasets, two ensembled datasets, and one machine learning dataset. In addition, we have added GLEAM and ERA5-Land ET dataset for comparison according to your valuable suggestions. We hope these revisions could address the importance of proposing the RSNP model for global ET estimation.

|                              | t t1        | 0         |                                 |                                                                                      |                                                     |
|------------------------------|-------------|-----------|---------------------------------|--------------------------------------------------------------------------------------|-----------------------------------------------------|
| Туре                         | ET Datasets | Time span | Spatial/Tempor
al resolution | Method                                                                               | Reference                                           |
| Remote
sensing
dataset | BESS        | 2001-2015 | 1km/8-day                       | Breathing Earth
System Simulator
process model                                 | (Jiang and
Ryu,
2016; Ryu
et al.,
2011) |
|                              | PML-V2      | 2002-2019 | 0.5 km/daily                    | PML model coupled
with gross primary
products via canopy
conductance theory | (Zhang et
al., 2019)                             |
|                              | PEW         | 1982-2018 | 0.1°/monthly                    | PT-JPL algorithm
considering available
water capacity                          | (Fu et al.,
2022)                                |
|                              | GLEAM4      | 1980-2023 | 0.1°/daily                      | GLEAM model                                                                          | (Miralles
et al.,
2025)                       |
|                              | VISEA       | 2001-2024 | 0.05°/daily                     | Variation of the
Standard
Evapotranspiration
Algorithm                      | (Huang et
al., 2024)                             |

| Table S1 | Summary | of typical | l global i | ЕΤ | datasets. |
|----------|---------|------------|------------|----|-----------|
|----------|---------|------------|------------|----|-----------|

|                                 | ETMonitor         | 2000-2021 | 1 km/monthly  | Estimating ET
components with a
multi-process
parameterization
model     | (Zheng et
al., 2022)             |
|---------------------------------|-------------------|-----------|---------------|--------------------------------------------------------------------------------------|-------------------------------------|
|                                 | 3T                | 2000–2020 | 0.25°/daily   | The Three-
temperature Mode                                                       | (Yu et al.,
2022)                |
|                                 | SSEBop            | 2003-now  | 1km/monthly   | Simplified Surface
Energy Balance model                                           | (Senay et al., 2020)                |
| Reanalysis
dataset           | ERA5-Land         | 1950-now  | 0.1°/monthly  | Hydrology-Tiled
ECMWF Scheme for
Surface Exchanges
over Land (H-
TESSEL) | (Muñoz-
Sabater et
al., 2021) |
|                                 | GLDAS             | 2000-now  | 0.25°/monthly | GLDAS NOAH Land
Surface model                                                     | (Rodell et al., 2004)               |
| Ensembled
datasets           | Synthesized
ET | 1982-2019 | 1km/monthly   | Ensemble the global
ET products                                                   | (Elnashar
et al.,
2021)       |
|                                 | GLASS             | 1982-2018 | 1km/8-day     | m/8-day Bayesian model
averaging method                                           |                                     |
| Machine
learning
datasets | FLUXCOM           | 2001-2015 | 0.1°/monthly  | Multiple machine learning methods                                                    | (Jung et
al., 2019)              |

Refences:

- Elnashar, A., Wang, L., Wu, B., Zhu, W., and Zeng, H.: Synthesis of global actual evapotranspiration from 1982 to 2019, Earth Syst. Sci. Data, 13, 447-480, http://doi.org/10.5194/essd-13-447-2021, 2021.
- Jiang, C. and Ryu, Y.: Multi-scale evaluation of global gross primary productivity and evapotranspiration products derived from Breathing Earth System Simulator (BESS), Remote Sens. Environ., 186, 528-547, https://doi.org/10.1016/j.rse.2016.08.030, 2016.
- Ryu, Y., Baldocchi, D. D., Kobayashi, H., Van Ingen, C., Li, J., Black, T. A., Beringer, J., Van Gorsel, E., Knohl, A., and Law, B. E.: Integration of MODIS land and atmosphere products with a coupled-process model to estimate gross primary productivity and evapotranspiration from 1 km to global scales, Global Biogeochem. Cycles, 25, https://doi.org/10.1029/2011GB004053, 2011.
- Zhang, Y., Kong, D., Gan, R., Chiew, F. H., McVicar, T. R., Zhang, Q., and Yang, Y.: Coupled estimation of 500 m and 8-day resolution global evapotranspiration and gross primary production in 2002-2017, Remote Sens. Environ., 222, 165-182, http://doi.org/10.1016/j.rse.2018.12.031, 2019.
- Fu, J., Wang, W., Shao, Q., Xing, W., Cao, M., Wei, J., Chen, Z., and Nie, W.: Improved global evapotranspiration estimates using proportionality hypothesis-based water balance constraints, Remote Sens. Environ., 279, 113140, http://doi.org/10.1016/j.rse.2022.113140, 2022.
- Miralles, D. G., Bonte, O., Koppa, A., Baez-Villanueva, O. M., Tronquo, E., Zhong, F., Beck, H. E., Hulsman, P., Dorigo, W., and Verhoest, N. E. C.: GLEAM4: global land evaporation and soil moisture dataset at 0.1 resolution from 1980 to near present, Sci. Data, 12, 1-14, https://doi.org/10.1038/s41597-025-04610-y, 2025
- Huang, L., Luo, Y., Chen, J. M., Tang, Q., Steenhuis, T., Cheng, W., and Shi, W.: Satellite-based

near-real-time global daily terrestrial evapotranspiration estimates, Earth Syst. Sci. Data Discuss., 2024, 1-37, https://doi.org/10.5194/essd-16-3993-2024, 2024.

- Zheng, C., Jia, L., and Hu, G.: Global land surface evapotranspiration monitoring by ETMonitor model driven by multi-source satellite earth observations, J. Hydrol., 613, 128444, https://doi.org/10.1016/j.jhydrol.2022.128444, 2022.
- Yu, L., Qiu, G. Y., Yan, C., Zhao, W., Zou, Z., Ding, J., Qin, L., and Xiong, Y.: A global terrestrial evapotranspiration product based on the three-temperature model with fewer input parameters and no calibration requirement, Earth Syst. Sci. Data Discuss., 2022, 1-33, https://doi.org/10.5194/essd-14-3673-2022, 2022.
- Senay, G. B., Kagone, S., and Velpuri, N. M.: Operational global actual evapotranspiration: Development, evaluation, and dissemination, Sensors, 20, 1915, https://doi.org/10.3390/s20071915, 2020.
- Muñoz-Sabater, J., Dutra, E., Agustí-Panareda, A., Albergel, C., Arduini, G., Balsamo, G., Boussetta,
  S., Choulga, M., Harrigan, S., and Hersbach, H.: ERA5-Land: A state-of-the-art global reanalysis dataset for land applications, Earth Syst. Sci. Data, 13, 4349-4383, 2021.
- Rodell, M., Famiglietti, J., Chen, J., Seneviratne, S., Viterbo, P., Holl, S., and Wilson, C.: Basin scale estimates of evapotranspiration using GRACE and other observations, Geophys. Res. Lett., 31, https://doi.org/10.1029/2004GL020873, 2004.
- Yao, Y., Liang, S., Li, X., Hong, Y., Fisher, J. B., Zhang, N., Chen, J., Cheng, J., Zhao, S., and Zhang, X.: Bayesian multimodel estimation of global terrestrial latent heat flux from eddy covariance, meteorological, and satellite observations, Journal of Geophysical Research: Atmospheres, 119, 4521-4545, https://doi.org/10.1002/2013JD020864, 2014.
- Jung, M., Koirala, S., Weber, U., Ichii, K., Gans, F., Camps-Valls, G., Papale, D., Schwalm, C., Tramontana, G., and Reichstein, M.: The FLUXCOM ensemble of global land-atmosphere energy fluxes, Sci. Data, 6, 74, https://doi.org/10.1038/s41597-019-0076-8, 2019.

4.) Unclear Justification for Chosen Comparison Datasets:

Following on from the previous comment, the study evaluates their ET dataset against three other global products, but the rationale for selecting these particular datasets is not provided. The omission of GLEAM, which is a widely used and well-validated ET dataset, is notable. The authors should justify their dataset choices of evaluation datasets —do they represent distinct modelling approaches or different data sources? Establishing a clear logic for dataset selection is necessary to ensure that the validation is robust and meaningful. A clear justification of the global ET dataset comparison would strengthen the study and make its need and value more obvious.

**Response:** We sincerely appreciate your insightful suggestion regarding the rationale for dataset selection. We have added the widely used and well-validated GLEAM ET product for comparison according to your suggestion, which could strengthen the robustness of our validation. The introduction of GLEAM ET datasets and its validation results have been added into the revised manuscript. And there are revisions we made with this comment. The scatter plot of validated sites at the site scale showed RSNP (R2=0.65, RMSE=23.19mm/month, bias=-3.81mm/month) has comparable accuracy with GLEAM (R2=0.66, RMSE=22.70mm/month, bias=-3.06mm/month). The validation results at the basin scale demonstrate that RSNP exhibits a higher coefficient of determination and lower error (R2=0.89, RMSE=113.04 mm/yr, RE=0.16) when compared to GLEAM (R2=0.84, RMSE=129.63 mm/yr, RE=0.17) in relation to water-balanced ET (WBET).

The results suggest that the RSNP ET dataset may offer greater accuracy than the GLEAM ET dataset for regional studies, and RSNP is response for regional hydrology studies. To further clarify our dataset selection, we note that the evaluated ET products represent distinct algorithmic approaches: "Among them, ETMonitor and PML-V2 developed from the PM algorithm, PEW and GLEAM are based on the Priestly-Taylor (PT) algorithm, and ERA5-Land is a reanalysis dataset derived from land surface model.". Additionally, in Section2.3.3, we have expanded on basic algorithms of each dataset to facilitate clearer evaluation. The varying spatial resolutions among the datasets also affect their comparability.

Figure 4: Comparison of estimated ET and observed ET over FLUXNET2015 sites. The relative mean square error (RMSE) and the bias are both in mm/month.

---

## Author Comment (AC4)

Response to Chaolei Zheng:

Dear reviewer, we sincerely appreciate your time and effort in reviewing our manuscript and providing valuable feedback to help improve our work. In the reply, the reviewer's comments are in black, our responses are in blue, and quotes from the revised manuscript are in *orange italics*.
* * *
I read the manuscript "A Globally Seamless Terrestrial Evapotranspiration Dataset Retrieved by a Nonparametric Approach with Remote Sensing and Reanalysis Datasets" with great interest. The new generated global ET data by RSNP model is a great contribution to the ET community. While the manuscript is generally well written and clear, I do have some specific comments and requests for clarification of the presented analyses.

ETMonitor ET dataset is seamless at daily resolution, and it even include open water evaporation and snow/ice sublimation in the terrestrial surface. I'm not sure exactly why the presented available ratio of pixels is low in some regions. It should be noted that extreme low ET value (e.g., zero) in ETMonitor product is valid, and the missing value is set as '-1' in the ETMonitor product. Please double check to make sure zero is not treated as unavailable data during the analysis.

**Response:** We sincerely appreciate your careful observation regarding the treatment of zero values in the ETMonitor ET dataset and we apologize for the error in our initial analysis about the ETMonitor ET dataset. ETMonitor ET dataset explicitly defines "-1" as missing data, while zero values over the global land surface are valid pixels. In the revised manuscript, we have corrected the statistical error by taking only '-1' as missing value in the ETMonitor product, and ETMonitor is seamless over the global land surface. The revised Figure 10 shows that ETMonitor has 100% available pixel ratios at the monthly scale, and the revised Fig.12 also shows ETMonitor's full spatial coverage. Thank you again for your valuable feedback, which has significantly improved the accuracy of our study.

Figure 11: The available pixel ratio of ET datasets at the monthly scale